# $\phi$-UPDATE: A CLASS OF POLICY UPDATE METHODS WITH POLICY CONVERGENCE GUARANTEE

**Wenye Li**[*], **Jiacai Liu**[*], **Ke Wei**[†]
School of Data Science, Fudan University

## ABSTRACT

Inspired by the similar update pattern of softmax natural policy gradient and Hadamard policy gradient, we propose to study a general policy update rule called $\phi$-update, where $\phi$ refers to a scaling function on advantage functions. Under very mild conditions on $\phi$, the global asymptotic state value convergence of $\phi$-update is firstly established. Then we show that the policy produced by $\phi$-update indeed converges, even when there are multiple optimal policies. This is in stark contrast to existing results where explicit regularizations are required to guarantee the convergence of the policy. Since softmax natural policy gradient is an instance of $\phi$-update, it provides an affirmative answer to the question whether the policy produced by softmax natural policy gradient converges. The exact asymptotic convergence rate of state values is further established based on the policy convergence. Lastly, we establish the global linear convergence of $\phi$-update.

## 1 INTRODUCTION

As a basis model, Markov Decision Process (MDP) has been widely studied and applied in reinforcement learning (RL). More precisely, an MDP model can be represented as a tuple $\mathcal{M}\left(\mathcal{S}, \mathcal{A}, P, r, \gamma\right)$, where $\mathcal{S}$ is the state space, $\mathcal{A}$ is the action space, $P$ is the state-transition model, $r : \mathcal{S} \times \mathcal{A} \to \mathbb{R}$ is the reward function, and $\gamma \in [0, 1)$ is the discounted factor. With a policy $\pi$, after selecting an action $a_t \sim \pi(\cdot|s_t)$ at a state $s_t$, the agent receives a reward $r_t = r(s_t, a_t)$ and then transfers to another state $s_{t+1}$ based on the transition model $P(\cdot|s_t, a_t)$. The state value function is defined as the expected discounted cumulative reward over random trajectories induced by $\pi$,

$$V^\pi(s) := \mathbb{E}\left[\sum_{t=0}^\infty \gamma^t r(s_t, a_t) \,\bigg|\, s_0 = s,\, \pi\right].$$

The target of RL is to find an optimal policy which maximizes the following objective function

$$\max_\pi\ V^\pi(\mu) \tag{1}$$

where $V^\pi(\mu) := \mathbb{E}_{s \sim \mu}\left[V^\pi(s)\right]$ for some initial state distribution $\mu$.

Policy optimization refers to a family of effective algorithms to solve (1). Compared with the classic dynamic programming approaches such as value iteration (VI) and policy iteration (PI) (Puterman, 1994; Sutton & Barto, 2018), policy optimization searches in the policy space based on policy parameterization. Policy gradient (PG (Sutton et al., 1999)) method is a basic policy optimization method which updates policy parameter by gradient ascent directly. Some other important variants of PG include natural policy

---

[*]Equal contribution.
[†]Corresponding author (Email: kewei@fudan.edu.cn).

gradient (NPG (Kakade, 2001)), policy mirror descent (PMD (Xiao, 2022)), trust region policy optimization (TRPO (Shani et al., 2020)), proximal policy optimization (PPO (Schulman et al., 2017)), and deep deterministic policy gradient (DDPG (Lillicrap et al., 2015)).

## 1.1 RELATED WORKS

Recently, the analysis of policy optimization methods has received intensive investigations, especially in the tabular setting. An exhaustive review towards this line of research is beyond the scope of this paper. Here we only summarize the results of the exact policy gradient methods that are mostly related to our work.

**Softmax PG** In Agarwal et al. (2021), the asymptotic global convergence of softmax PG with sufficiently small step-size is established, which is generalized to the case of any positive step-size in Liu et al. (2024b). For non-asymptotic error bounds, the $O(1/k)$ sublinear upper and lower bounds under small constant step-size are provided in Mei et al. (2020b) by leveraging the smoothness and the gradient domination property of the value function. These results are also extended to an arbitrary step-size in Liu et al. (2024b). In addition, the linear convergence of softmax PG under adaptive step-size is established in Mei et al. (2021); Liu et al. (2024b).

**Escort PG and Hadamard PG** Escort PG is proposed in Mei et al. (2020a) to overcome the limitation of softmax PG. It is shown that with some adaptive step-size, escort PG converges at the rate of $O(1/k)$ which enjoys a better constant dependence of $c^{2-2/p}$ than $c^2$ for softmax PG (Mei et al., 2020b), where $c < 1$ can be extremely small. As a specific instance of escort PG, Hadamard PG is studied in Liu et al. (2023) and proved to have a local linear convergence.

**Softmax NPG and PMD** As shown in Agarwal et al. (2021), softmax NPG enjoys a dimension-free $O(1/k)$ sublinear convergence rate. The local linear convergence rate of softmax NPG is established in Khodadadian et al. (2021) by leveraging the contraction of non-optimal probability. The global linear convergence is established for geometrically increasing step-sizes in Xiao (2022) and for adaptive step-sizes in Khodadadian et al. (2021); Johnson et al. (2023). In Liu et al. (2024b), the global linear convergence of softmax NPG under any constant step-size is established.

PMD can be viewed as an extension of softmax NPG. The $O(1/k)$ sublinear convergence rate of softmax NPG in Agarwal et al. (2021) is generalized to PMD in Xiao (2022), while the global linear convergence with geometrically increasing step-sizes is also provided therein. PMD with regularization has been studied in Lan (2021); Li et al. (2023b); Zhan et al. (2023), while PMD with function approximations has been provided in Tomar et al. (2020); Yuan et al. (2023); Alfano et al. (2023).

## 1.2 MOTIVATION AND CONTRIBUTION

As already mentioned, softmax PG admits a strict $O(1/k)$ sublinear convergence rate. In contrast, Hadamard PG and softmax NPG can break this limitation and achieve linear convergence. The policy update rules of these algorithms are listed below:

$$\begin{aligned}
\text{(Softmax PG)} \quad & \pi^+(a|s) \propto \pi(a|s) \cdot \exp\left(\frac{\eta}{1-\gamma} d_\mu^\pi(s) \pi(a|s) A^\pi(s,a)\right), \\
\text{(Softmax NPG)} \quad & \pi^+(a|s) \propto \pi(a|s) \cdot \exp\left(\eta A^\pi(s,a)\right), \\
\text{(Hadamard PG)} \quad & \pi^+(a|s) \propto \pi(a|s) \cdot \left(1 + \frac{2\eta}{1-\gamma} d_\mu^\pi(s) A^\pi(s,a)\right)^2,
\end{aligned}$$

where $\pi^+$ is the updated policy, $\eta$ is the step-size, the *visitation measure* $d_\mu^\pi$ is defined as

$$\forall s \in \mathcal{S}: \quad d_\mu^\pi(s) := (1-\gamma) \sum_{t=0}^\infty \gamma^t \Pr\left(s_t = s | s_0 \sim \mu, \pi, P\right),$$

the *advantage function* $A^\pi$ is defined as

$$A^\pi(s, a) := Q^\pi(s, a) - V^\pi(s),$$

and $Q^\pi(s, a) := r(s, a) + \gamma \mathbb{E}_{s' \sim P(\cdot|s,a)} [V^\pi(s')]$ is the state-action value function. From the update rules above, it can be observed that the policy update direction of softmax PG is influenced by the term $\pi(a|s)$, which slows down the convergence rate as shown in Li et al. (2023a). By contrast, the policy update directions of both softmax NPG and Hadamard PG are aligned with $A^\pi(s, a)$, which motivates us to study a class of policy update methods called $\phi$-update, taking the form of

$$(\phi\text{-update}) \quad \forall s \in \mathcal{S}, \ a \in \mathcal{A} : \quad \pi^+(a|s) \propto \pi(a|s) \cdot \phi\left(\eta_s^\pi A^\pi(s, a)\right).$$

Here $\eta_s^\pi$ is the step-size that can vary with the policy $\pi$ and state $s$, and $\phi$ is some real-valued function.

The goal of this paper is to conduct a systematic convergence analysis of $\phi$-update given the access to exact policy gradients under the tabular setting. The main contributions of this paper are summarized as below:

**Global convergence of $\phi$-update**  We first show that $\phi$-update converges globally to the optimal value under very mild conditions on $\phi$ (Theorem 3.1). It is noted that the existing global convergence analysis is not applicable for $\phi$-update. For instance, the global convergence analysis of sofmax PG in Agarwal et al. (2021) and Hadamard PG in Liu et al. (2023) explicitly utilizes the policy parameterization which is not available for $\phi$-update. In addition, the global convergence analysis of NPG and PMD in Xiao (2022) relies on the three point descent lemma while $\phi$-update does not enjoy such property in general. Therefore, new techniques have been developed.

**Policy convergence and exact asymptotic convergence rate**  A key contribution of this paper is the establishment of the policy convergence of $\phi$-update even when there are multiple optimal policies (Theorem 3.3). As an instance of $\phi$-update, this implies that softmax NPG (without any regularizations) converges in the policy domain, which is a new result for softmax NPG. To the best of our knowledge, this is the first policy convergence result for vanilla policy gradient methods without regularizations. Based on the convergence of the policy, the exact asymptotic convergence rate of state values is further established (Theorem 3.4), which is also applicable for softmax NPG.

**Global linear convergence**  By combining the local linear convergence result (Theorem 3.2) and the dynamic convergence result (Theorem 3.5), the global linear convergence of $\phi$-update is also established (Theorem 3.6).

## 2 NOTATIONS, SETTINGS AND $\phi$-UPDATE

### 2.1 NOTATIONS AND SETTINGS

Recall that MDP is represented as a tuple $\mathcal{M}(\mathcal{S}, \mathcal{A}, P, r, \gamma)$. In this paper, we focus on the finite MDPs, i.e., $|\mathcal{S}|, |\mathcal{A}| < \infty$. Without loss of generality, we assume the reward function is bounded in $[0, 1]$, i.e., $r(s, a) \in [0, 1]$ for all $s$ and $a$.

Let $\Delta(\mathcal{A})$ be the probabilistic simplex on $\mathcal{A}$. Given a policy $\pi : \mathcal{S} \to \Delta(\mathcal{A})$, the value function $V^\pi(s)$, action value function $Q^\pi(s, a)$, and advantage function $A^\pi(s, a)$ are defined above. In this work, we consider the setting where the exact policy evaluation can be accessed. When a policy sequence $\{\pi^k\}$ is given, we use $V^k$, $Q^k$, and $A^k$ to represent $V^{\pi^k}$, $Q^{\pi^k}$, and $A^{\pi^k}$, respectively.

We let $\mathbb{1}\{\cdot\}$ be the indicator function. For an arbitrary vector $V \in \mathbb{R}^{|\mathcal{S}|}$, the Bellman operator induced by $\pi$ is defined as

$$\mathcal{T}^\pi V(s) := \mathbb{E}_{a \sim \pi(\cdot|s)} \mathbb{E}_{s' \sim P(\cdot|s,a)} [r(s, a) + \gamma V(s')].$$

As is common in the analysis of policy optimization methods, we assume the initial distribution $\mu$ is bounded away from zero (i.e., $\tilde{\mu} := \min_{s \in \mathcal{S}} \mu(s) > 0$). It follows immediately that $d_\mu^\pi(s) > 0$ for all $s \in \mathcal{S}$.

It is well known that there exists an optimal policy which maximizes $V^\pi(s)$ (e.g. in Puterman (1994)), denoted $\pi^*$ (may be non-unique). The corresponding optimal value function is denoted as $V^*$. The optimal action and advantage values $Q^*$ and $A^*$ are similarly defined.

Define the $\pi$-optimal action set at state $s$,

$$\mathcal{A}_s^\pi := \arg\max_a A^\pi(s, a),$$

and let $\pi_s(\mathcal{A}_s^\pi) := \sum_{a \in \mathcal{A}_s^\pi} \pi(a|s)$. Similarly, when a policy sequence is given we use $\mathcal{A}_s^k$ to represent $\mathcal{A}_s^{\pi^k}$. When $\pi = \pi^*$, we let $\mathcal{A}_s^* := \mathcal{A}_s^{\pi^*}$ be the optimal action set at state $s$. Let $\Delta$ be the optimal advantage function gap. That is,

$$\Delta := \min_{s \in \tilde{\mathcal{S}}, a \notin \mathcal{A}_s^*} |A^*(s, a)|,$$

where $\tilde{\mathcal{S}} := \{s \in \mathcal{S} : \mathcal{A}_s^* \neq \mathcal{A}\}$ is assumed to be non-empty.

## 2.2 $\phi$-UPDATE AND ASSUMPTIONS

In this section, we give the formal definition of $\phi$-update, as well as the assumptions required in the convergence analysis.

**Definition 2.1** ($\phi$-update). *Let $\phi : (-L, L) \to \mathbb{R}$ be a continuous scaling function, where $0 < L \leq +\infty$. In the $k$-th iteration, given step-size $\eta_s^k > 0$ such that $\left| \eta_s^k A^k(s, a) \right| < L$, the $\phi$-update takes the form of*

$$\forall k \in \mathbb{N}, \ s \in \mathcal{S}, \ a \in \mathcal{A} : \quad \pi^{k+1}(a|s) = \frac{\pi^k(a|s) \cdot \phi\left(\eta_s^k A^k(s, a)\right)}{Z_s^k},$$

*where $Z_s^k = \sum_a \pi^k(a|s) \cdot \phi\left(\eta_s^k A^k(s, a)\right)$ is the normalization factor.*

Without further specification, we assume $\pi^0(a|s) > 0, \forall s \in \mathcal{S}, a \in \mathcal{A}$ throughout this paper. Then it is easy to verify that there holds $\pi^k(a|s) > 0$ for any finite time $k$.

**Assumptions.** The following assumptions will be made on $\phi$:

(I) $\phi$ is strictly monotonically increasing, i.e., $\phi(x) > \phi(y)$ for all $x > y$,

(II) $\phi$ is strictly positive, i.e., $\phi(x) > 0$ for all $x$,

(III) $\phi$ is differentiable around $0$ and there exists some small $\delta > 0$ and $0 < c < +\infty$ such that

$$\frac{d}{dt} \log \phi(t) = \frac{\phi'(t)}{\phi(t)} \leq c$$

holds when $t \in (-\delta, \delta)$.

The first two assumptions are sufficient for us to establish the global asymptotic convergence and global linear convergence of $\phi$-update, but we further need the third one to establish the policy convergence and the exact asymptotic convergence rate. These three assumptions are indeed very mild and can even be satisfied by non-convex functions such as $\phi(t) = (1 + \exp(-t))^{-1}$ (sigmoid function) and $\phi(t) = \tan(t) + 1$ satisfy all the three assumptions. In particular, softmax NPG is an instance of $\phi$-update with $\phi(t) = \exp(t)$ and $L = +\infty$, and Hadamard PG is is an instance of $\phi$-update with $\phi(t) = (1 + 2t)^2$, $L = 1/2$.

## 3 CONVERGENCE ANALYSIS OF $\phi$-UPDATE

In this section we give a detailed analysis of $\phi$-update. The main results and their relations are summarized in Figure 1. We would like to emphasize again, in addition to the common convergence results such as the global asymptotic convergence and local and global linear convergence, the policy convergence (without any regularizations) and the exact asymptotic convergence rate have been established in this paper. The proof details of this section are presented in Appendices C, D and E. For simplicity, here we only present the convergence results under the constant step size, i.e. $\eta_s^k = \eta > 0$. It is worth noting that the convergence results can be easily generalized to adaptive step sizes, with related results being presented in Appendix H.

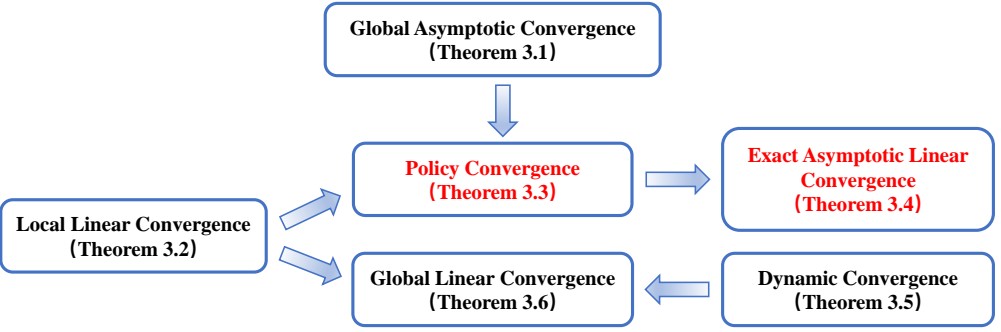

Figure 1: Main convergence results of $\phi$-update and their relations.

### 3.1 GLOBAL ASYMPTOTIC CONVERGENCE

The key ingredient in the analysis is the *improvement* at each state $s$ in each iteration $k$, represented as

$$\mathcal{T}^{k+1}V^k(s) - V^k(s) = \sum_a \pi^{k+1}(a|s)A^k(s,a).$$

We start from the following ascent property of $\phi$-update. It can be proved by a direct computation, see Appendix C.1.

**Proposition 3.1.** *Suppose the assumptions* (I), (II) *hold. For any positive step-size* $\eta > 0$, *$\phi$-update has non-negative improvement at every state, i.e.,* $\sum_a \pi^{k+1}(a|s)A^k(s,a) \geq 0$ *for any $s$ and $k$. Furthermore, we have* $V^{k+1}(s) \geq V^k(s)$ *for every* $s \in \mathcal{S}$.

As the value function is bounded (Lemma A.1 in the appendix), Proposition 3.1 implies that $V^\infty(s) := \lim_{k \to \infty} V^k(s)$ exists for all $s \in \mathcal{S}$. The following theorem establishes the global convergence of $\phi$-update, i.e. $V^\infty(s) = V^*(s), \forall s \in \mathcal{S}$.

**Theorem 3.1** (Global Asymptotic Convergence). *Suppose the assumptions* (I), (II) *hold. For a positive step-size* $\eta > 0$, *the value function generated by $\phi$-update method converges to the optimal value, i.e.,* $V^\infty(s) = V^*(s)$ *for all the states.*

**Remark 3.1.** *It is worth noting that under the softmax policy parameterization, $\phi$-update can be formulated as* $\theta_{s,a}^{k+1} = \theta_{s,a}^k + \log \phi(\eta_s^k A^k(s,a))$. *Despite this, the proof carried out in the parameter space in Agarwal et al. (2021) cannot be applied to $\phi$-update with this parameter formulation. One of the key ingredients therein for the proof is the property* $\sum_a \theta_{s,a}^k = \sum_a \theta_{s,a}^0, \forall k$, *which comes from the fact* $\sum_a \frac{\partial V^\pi(s)}{\partial \theta_{s,a}} = 0$ *and*

*the parameter update form of $\theta_{s,a}^{k+1} = \theta_{s,a}^k + \eta \frac{\partial V^k(s)}{\partial \theta_{s,a}^k}$. In general, such property is not satisfied for $\phi$-update under the same softmax parameterization. Instead, our proof is conducted completely in the policy space.*

As already mentioned, the existing techniques (e.g. Agarwal et al., 2021; Liu et al., 2023; Xiao, 2022) are not applicable since we only require very few assumptions on $\phi$ and there are no particular structures to use. In contrast, our proof leverages the explicit policy update formula of $\phi$-update to discuss the behavior of the policy on different actions and compute the improvement. A contradiction is further constructed by showing that the improvement is strictly positive based on the statistical decoupling technique. See Appendix C.2 for the proof details.

## 3.2 POLICY CONVERGENCE AND EXACT ASYMPTOTIC LINEAR CONVERGENCE

Here we first establish the policy convergence of the $\phi$-update, and then present the exact asymptotic convergence rate. To this end, we need to first study the local convergence of $\phi$-update. Note that, by the global convergence result (Theorem 3.1), for an arbitrarily small $\varepsilon > 0$, there exists a time $T(\varepsilon)$ such that

$$\forall \, k \geq T(\varepsilon): \quad \left\| V^* - V^k \right\|_\infty \leq \varepsilon.$$

In such an $\varepsilon$-sub-optimal region, we establish the following local linear convergence.

**Theorem 3.2** (Local Linear Convergence). *Suppose the assumptions* (I), (II) *hold. Define*

$$\rho_1(\eta, \Delta, \varepsilon) = \frac{\phi\left(-\eta(\Delta - \varepsilon)\right)}{\phi\left(-\eta\varepsilon\right)} \cdot \left(1 - \frac{\varepsilon}{\Delta}\right)^{-1}.$$

*Let $\varepsilon > 0$ be a constant such that $\rho_1(\eta, \Delta, \varepsilon) < 1$. Then the values generated by $\phi$-update method satisfy*

$$\forall \, k \geq T(\varepsilon): \quad \left\| V^* - V^k \right\|_\infty \leq \frac{\rho_1(\eta, \Delta, \varepsilon)^{(k - T(\varepsilon))}}{(1 - \gamma)^2 \tilde{\mu}^2}.$$

The proof is inspired by Khodadadian et al. (2021), which considers the policy ratio on sub-optimal actions. Lemma B.1 in Appendix B shows that $\pi^{k+1}(a'|s)/\pi^k(a'|s) \leq \rho_1(\eta, \Delta_s, \varepsilon)$. Combining it with the performance difference lemma (Lemma A.4 in Appendix A) gives the result. See Appendix D.1 for the proof details.

Based on the global asymptotic convergence (Theorem 3.1) and the local linear convergence (Theorem 3.2), we can establish the convergence of $\phi$-update in the policy domain with an additional mild condition on $\phi$.

**Theorem 3.3** (Policy Convergence). *Suppose the assumptions* (I), (II), (III) *hold. Then the policy generated by $\phi$-update converges to some optimal policy $\pi^*$, i.e. the sequence $\{\pi^k(a|s)\}_k$ converges to some $\pi^*(a|s)$ for any $s \in \mathcal{S}$ and $a \in \mathcal{A}$.*

It suffices to show that $\pi^k(a^*|s)$ converges for all the optimal actions $a^* \in \mathcal{A}_s^*$, as $\pi^k(a'|s)$ vanishes for all $a \notin \mathcal{A}_s^*$ by the global convergence result (Theorem 3.1). This can be achieved by showing that $\left\{\log \pi^k(a^*|s)\right\}_k$ is a Cauchy sequence. To this end, we first bound the policy ratio on the optimal actions in Lemma B.2 (see Appendix B). Further utilizing the local linear convergence result (Theorem 3.2) can show that $\sum_{k=T(\varepsilon)}^\infty \left|\log \pi^{k+1}(a^*|s) - \log \pi^k(a^*|s)\right| \lesssim \varepsilon$. See Appendix D.2 for the proof details.

**Remark 3.2.** *For softmax NPG, one has $\phi(t) = \exp(t)$. Thus, it is easy to see that $\phi$ satisfies the assumptions* (I), (II), (III). *In addition, there holds*

$$\frac{\phi'(t)}{\phi(t)} \equiv 1, \quad \forall \, t \in \mathbb{R}.$$

**Remark 3.3.** *In Li et al. (2023b), it is shown that the policy produced by the homotopic policy mirror descent method converges to a uniform policy when there are multiple optimal policies. However, it should be noted that the analysis therein relies on the explicit entropy regularization in the algorithm. For vanilla policy gradient methods without regularizations, Theorem 3.3 is the first policy convergence guarantee result, to the best of our knowledge.*

Based on the convergence of the policy, the following result can be further established.

**Theorem 3.4** (Exact Asymptotic Linear Convergence). *Suppose the assumptions* (I), (II), (III) *hold. Then one has*

$$\lim_{k \to \infty} \frac{V^*(\mu) - V^{k+1}(\mu)}{V^*(\mu) - V^k(\mu)} = \frac{\phi(-\eta\Delta)}{\phi(0)}.$$

The key of the proof is the observation that the policy probability on non-optimal actions concentrates on the action set $\mathcal{A}'_{s_0}$ where $s_0 \in \arg\min_s \Delta_s$, and the proof is to verify that the value error ratio is asymptotically equal to the policy ratio on $\mathcal{A}'_{s_0}$. See Appendix D.3 for the proof details.

**Remark 3.4.** *It can be observed that the exact asymptotic rate does not depend on $\gamma$ explicitly (but $\gamma$ may affect $\Delta$). To interpret it intuitively, first note that we have $A^k \approx A^*$ when the policy generated by $\phi$-update is nearly optimal. In such case, the problem can be roughly viewed as a bandit problem. That is, at each state, we just need to find the actions with the largest advantage, i.e. $\arg\max_a A^*(s, a)$. As the bandit problem is an MDP with $\gamma = 0$, it is natural that the asymptotic rate does not explicitly rely on $\gamma$.*

**Remark 3.5.** *To obtain a more intuitive understanding of $\phi(-\eta\Delta)/\phi(0)$, consider a bandit problem (i.e. only one state $s$ and $\gamma = 0$) where there are only two actions $a_1$, $a_2$. Assume $r(s, a_1) = r > 0$, $r(s, a_2) = 0$. Then it is clear that the optimal action is $a_1$, the non-optimal action is $a_2$, and $\Delta = r$. By the performance difference lemma and the explicit formula of $\phi$-update there holds*

$$\frac{V^*(\mu) - V^{k+1}(\mu)}{V^*(\mu) - V^k(\mu)} = \frac{\pi^{k+1}(a_2|s)}{\pi^k(a_2|s)} = \frac{\phi\left(\eta A^k(s, a_2)\right)}{\pi^k(a_1|s)\phi\left(\eta A^k(s, a_1)\right) + \pi^k(a_2|s)\phi\left(\eta A^k(s, a_2)\right)}.$$

*By the global convergence (Theorem 3.1), we have $\pi^k(a_1|s) \to 1$, $\pi^k(a_2|s) \to 0$, $A^k(s, a_1) \to A^*(s, a_1) = 0$ and $A^k(s, a_2) \to A^*(s, a_2) = -\Delta$. Hence the asymptotic rate $\phi(-\eta\Delta)/\phi(0)$ is observed.*

**Remark 3.6.** *Theorem 3.4 shows that the exact asymptotic convergence rate of softmax NPG is $\exp(-\eta\Delta)$, which is also a new result for softmax NPG. Note that the $O(e^{-(1-1/\lambda)\eta\Delta})$ upper bound and and the $O(e^{-\eta(\Delta+\varepsilon)})$ lower bound for local linear convergence of softmax NPG have been established in Khodadadian et al. (2021) and Liu et al. (2024b) respectively, where $\lambda$ and $\varepsilon$ are certain constants. It is worth noting that the exact asymptotic rate in this paper cannot be obtained directly from these two local bounds, but requires a different technique for proof.*

### 3.3 GLOBAL LINEAR CONVERGENCE

Finally, we present the global linear convergence of the value function generated by $\phi$-update. The overall analysis idea is inspired by Liu et al. (2024b). First the following lemma in Liu et al. (2024b) shows linear contraction can be achieved when the improvement has an $\Omega(\max_a A^k(s, a))$ lower bound.

**Lemma 3.1** (Lemma 2.12 (Liu et al., 2024b)). *Assume $\sum_a \pi^{k+1}(a|s)A^k(s, a) \geq C_k \max_a A^k(s, a)$ holds for some $C_k \in [0, 1]$ and all $s \in \mathcal{S}$, then*

$$\left\|V^* - V^{k+1}\right\|_\infty \leq (1 - (1 - \gamma)C_k)\left\|V^* - V^k\right\|_\infty.$$

The following improvement lower bound can be established for $\phi$-update.

**Lemma 3.2** (Improvement Lower Bound). *Suppose the assumptions* (I), (II) *hold. With positive step-size* $\eta > 0$, *the $\phi$-update improvement of state $s$ satisfies*

$$\sum_a \pi^{k+1}(a|s)A^k(s,a) \geq \left[1 - \frac{1}{1 + \pi_s^k\left(\mathcal{A}_s^k\right)\left(\Delta_{\phi,s}^k(\eta) - 1\right)}\right] \cdot \max_a A^k(s,a),$$

*where*

$$\Delta_{\phi,s}^k(\eta) := \frac{\phi\left(\eta \max_a A^k(s,a)\right)}{\mathbb{E}_{a' \sim \xi^k(\cdot|s)}\left[\phi\left(\eta A^k(s,a')\right)\right]} \quad \text{with } \xi^k(a|s) = \begin{cases} 0 & \text{if } a \in \mathcal{A}_s^k, \\ \pi^k(a|s)/(1 - \pi_s^k(\mathcal{A}_s^k)) & \text{if } a \notin \mathcal{A}_s^k. \end{cases}$$

This lower bound is established by a direct computation. See Appendix E.2 for the proof details.

**Remark 3.7.** *The $\Delta_{\phi,s}^k$ can be seemed as a dynamic measure of the $\phi$-scaled advantage function gap at $k$-th iteration, as $\xi^k$ is the policy value conditioned on the non-optimal action set.*

Together with Lemma 3.1, the following dynamic convergence result of $\phi$-update can be obtained directly. See Appendix E.3 for the proof.

**Theorem 3.5** (Global Dynamic Convergence). *Suppose the assumptions* (I), (II) *hold. With positive step-size* $\eta > 0$, *the value function generated by $\phi$-update satisfies*

$$\forall k \in \mathbb{N}^+ : \quad \left\|V^* - V^k\right\|_\infty \leq \left\|V^* - V^0\right\|_\infty \prod_{t=0}^{k-1}\left(1 - (1-\gamma)\left[1 - \frac{1}{1+D_t}\right]\right),$$

*where $D_t := \min_{s \in \tilde{\mathcal{S}}_t}\left\{\pi_s^t(\mathcal{A}_s^t)\left(\Delta_{\phi,s}^t(\eta) - 1\right)\right\}$ and $\tilde{\mathcal{S}}_t := \{s \in \mathcal{S} : \mathcal{A}_s^t \neq \mathcal{A}\}$.*

**Remark 3.8.** *There exists a bandit example which shows the error bound in Theorem 3.5 is tight, see Appendix E.4. Note that $D_t$ is likely to be very small. Indeed, it is interesting to see whether there is a worst case example which shows $D_t$ can be exponentially small, as in Li et al. (2023a). It is also interesting to see whether we can use the techniques in a recent work (Klein et al., 2024) to address this issue.*

We are now ready to establish the global linear convergence of $\phi$-update by combining Theorem 3.2 and Theorem 3.5. See Appendix E.5 for the proof details.

**Theorem 3.6** (Global Linear Convergence). *Suppose the assumptions* (I), (II) *hold. Define*

$$\rho(\varepsilon) = \max\left\{\kappa(\varepsilon),\ \rho_1(\eta, \Delta, \varepsilon)\right\}, \quad \text{where } \kappa(\varepsilon) = \max_{t \leq T(\varepsilon)}\left\{1 - (1-\gamma)\left[1 - \frac{1}{1+D_t}\right]\right\}.$$

*There exists $\varepsilon > 0$ such that $\rho(\varepsilon) < 1$ and*

$$\forall k \in \mathbb{N}^+ : \quad \left\|V^* - V^k\right\|_\infty \leq \frac{\left\|V^* - V^0\right\|_\infty}{(1-\gamma)\tilde{\mu}^2} \cdot \rho(\varepsilon)^k.$$

## 4 EMPIRICAL VALIDATIONS AND DISCUSSION

### 4.1 EMPIRICAL VALIDATIONS

We empirically validate Theorems 3.3 and 3.4 using $\phi(t) = \exp(t)$ (corresponding to softmax NPG), $\phi(t) = (1 + \exp(-t))^{-1}$, and $\phi(t) = \tan(t) + 1$. The step size is set to $\eta = 1$, $\eta = 1$, and $\eta = 0.1$, respectively. It can be easily verified the last two scaling functions also satisfy the assumptions (I), (II), (III).

For Theorem 3.3, we use a simple $5 \times 5$ Grid World problem to test the policy convergence of $\phi$-update. The computational results are presented in Figure 2. The policy corresponding to the last iterate of $\phi$-update (denoted $\pi^{\text{last}}$) is presented in the second row of Figure 2. The plots in the first row display $\left\|\pi^k - \pi^{\text{last}}\right\|_1$ against the number of iterations which clearly show the convergence of the policy for the three tested $\phi$.

For Theorem 3.4, we use the random MDP to valid the exact asymptotic rate. The computational results are presented in Figure 3. It can be observed from the plots that the theoretical asymptotic rate match the empirical one very well.

In addition, more examples of $\phi$-update have been tested, including extensions of softmax NPG and a family of polynomial update, see Appendix F.

### 4.2 IMPLEMENTATION UNDER GENERAL POLICY PARAMETERIZATIONS

Though we focus on the convergence analysis of $\phi$-update given the access to exact policy gradients in the tabular setting, it is worth pointing out that $\phi$-update can be implemented under general policy parameterizations (e.g., artificial neural networks). As inspired by Tomar et al. (2020), we provide an overall idea of the supervised learning implementation. More precisely, given a distribution over state space $\mathcal{S}$ and policy parameters $\theta^k$ (or equivalently the policy $\pi^k$) in the $k$-th iteration, denote by $\pi_\phi^k$ the target policy generated by $\phi$-update given $\pi^k$ and $\eta_k$. One can solve for the new parameters $\theta^{k+1}$ by minimizing the KL divergence with $\pi_\phi^k$:

$$\theta^{k+1} \in \arg\min_\theta \mathbb{E}_{s \sim \nu} \left[ \text{KL} \left( \pi_\theta \left( \cdot | s \right) \,||\, \pi_\phi^k \left( \cdot | s \right) \right) \right]$$

$$= \arg\min_\theta \mathbb{E}_{s \sim \nu} \left[ \mathbb{E}_{a \sim \pi^k(\cdot|s)} \left[ \frac{\pi_\theta \left( a | s \right)}{\pi^k \left( a | s \right)} \left[ \log \frac{\pi_\theta \left( a | s \right)}{\pi^k \left( a | s \right)} - \log \phi \left( \eta_k A^k \left( s, a \right) \right) \right] \right] \right]. \tag{2}$$

It is worth noting that the normalization factor of $\phi$-update is not an issue in this supervised learning framework.

This idea can be implemented in both on-policy and off-policy scenarios for real problems. For instance, one can use a critic network with parameter $\omega$ to fit the value function. Collecting on-policy rollouts $\mathcal{D} = \left\{ (s_i, a_i, r_i, s_i') \right\}_{i=1}^n$ from current policy $\pi^k$, methods such as GAE (Schulman et al., 2020) can be performed to estimate the advantages $A^k$ and the value targets $V_{\text{targ}}^k$. Then the actor parameter $\theta$ and critic parameter $\omega$ are updated simultaneously by optimizing the policy objective loss (Eq (2)) plus a value loss term. It is also feasible to extend this procedure to the off-policy scenario by collecting reply buffer $\mathcal{D}$, updating the critic parameters $\omega$ with methods like Temporal-Difference learning (see Sutton & Barto (2018) for instance) and optimizing Eq (2) to update the actor parameter $\theta$ with the estimated advantages. Some preliminary results about the implementation of $\phi$-update under the neural network parameterization have been reported in Appendix G.

## 5 CONCLUSION AND FUTURE WORK

In this paper, we have investigated a class of policy optimization methods called $\phi$-update. For a wide family of scaling functions, we have presented a series of convergence results for $\phi$-update. In particular, the convergence of the policy is established even when the optimal policy is not unique. The exact asymptotic linear convergence rate is further established based on this result.

It should be noted that this work has been primarily concerned with the case of exact policy evaluation. As most of the RL applications are under stochastic settings, the generalization to the analysis of inexact $\phi$-update is one of the future works. Additionally, it is interesting to seek for more efficient $\phi$ in practice and implement $\phi$-update for real problems via deep neural networks.

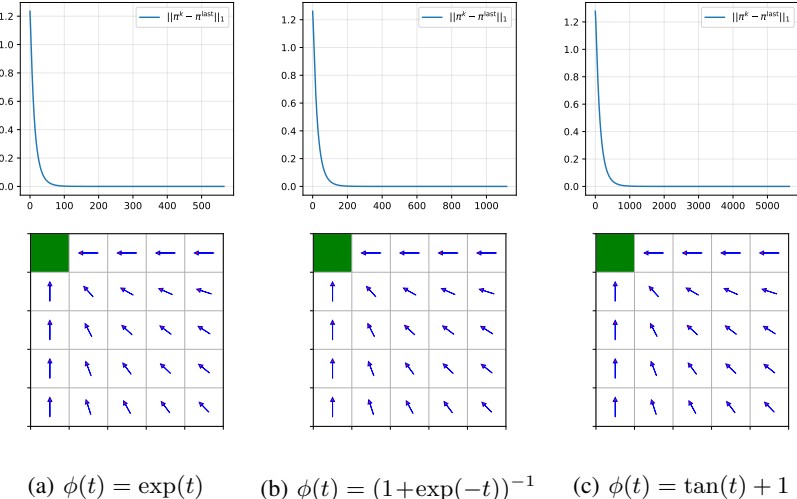

(a) $\phi(t) = \exp(t)$      (b) $\phi(t) = (1+\exp(-t))^{-1}$      (c) $\phi(t) = \tan(t) + 1$

Figure 2: The simulation results on the a simple $5 \times 5$ Grid World to test policy convergence. For each grid except the green one, the agent can choose five actions (up, down, left, right, and stay) with no rewards. The green grid is the terminal state. Once the agent steps in, it receives a $+1$ reward and the game is terminated. The discount factor $\gamma$ is set to $0.9$, and we use a uniform policy as the initial policy. Note that there exists multiple optimal policies for this Grid World problem. Indeed, except the first row and the first column, there are two optimal actions for each grid (up and left). The $\phi$-update is run for a sufficiently number of iterations such that $V^*(\rho) - V^k(\rho) \leq 10^{-16}$ ($\rho$ is the uniform distribution over $\mathcal{S}$), and the policy corresponding to the last iterate (denoted $\pi^{\text{last}}$) is presented in the second row. The oblique arrows in the plot essentially mean that the optimal policy produced by the $\phi$-update has non-zero probability on every optimal action. The plots in the first row display $\left\|\pi^k - \pi^{\text{last}}\right\|_1$ against the number of iterations.

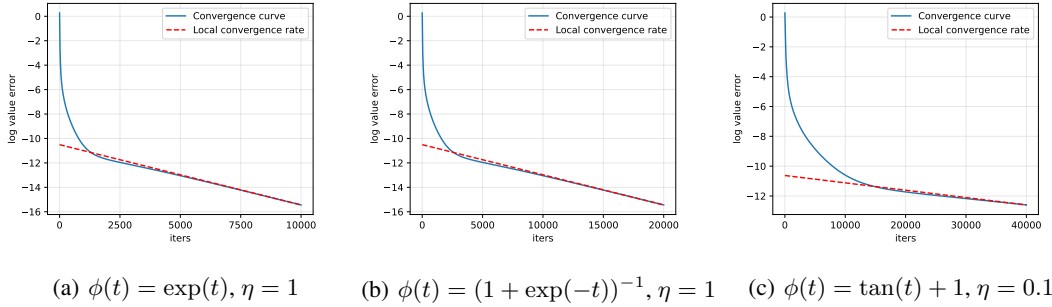

(a) $\phi(t) = \exp(t), \eta = 1$      (b) $\phi(t) = (1 + \exp(-t))^{-1}, \eta = 1$      (c) $\phi(t) = \tan(t) + 1, \eta = 0.1$

Figure 3: The simulation results on the random MDP with $|\mathcal{S}| = 50$, $|\mathcal{A}| = 10$ and $\gamma = 0.7$. The reward $r(s, a)$ and transition probability $P(s'|s, a)$ are uniformly generated from $[0, 1]$ ($P$ is further normalized to be a probability matrix). The initial state distribution $\mu$ is uniform on $\mathcal{S}$. Blue curve is the log-value-error and the red line corresponds to the theoretical convergence rate in Theorem 3.4.

ACKNOWLEDGEMENTS

Ke Wei was partially supported by the National Key R&D Program of China (No. 2023YFA1009300), National Natural Science Foundation of China (No. 92370105), Natural Science Foundation of Shanghai (No. 23ZR1406400), and Science and Technology Commission of Shanghai Municipality (No. 23JC1401000).

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

# Appendix

## A  USEFUL LEMMAS

In this section we introduce several lemmas which are useful in our further analysis. The proofs are omitted as these lemmas can be verified easily or can be found in previous works.

**Lemma A.1.** *Under the assumption of $r(s, a) \in [0, 1]$, for arbitrary $\pi \in \Pi$, $s \in \mathcal{S}$. $a \in \mathcal{A}$ we have*

$$V^\pi(s) \in \left[0, \ \frac{1}{1-\gamma}\right], \quad Q^\pi(s,a) \in \left[0, \ \frac{1}{1-\gamma}\right], \quad A^\pi(s,a) \in \left[-\frac{1}{1-\gamma}, \ \frac{1}{1-\gamma}\right].$$

**Lemma A.2.** *For any policy $\pi$:*

$$\left\|A^* - A^\pi\right\|_\infty \leq \left\|V^* - V^\pi\right\|_\infty.$$

In this paper we use two metrics for the value error, $V^*(\mu) - V^\pi(\mu)$ and $\left\|V^* - V^\pi\right\|_\infty$. The following lemma says that these two metrics can be bounded by each other.

**Lemma A.3.** *For any policy $\pi$,*

$$\tilde{\mu} \cdot \left\|V^* - V^\pi\right\|_\infty \leq V^*(\mu) - V^\pi(\mu) \leq \left\|V^* - V^\pi\right\|_\infty.$$

**Lemma A.4** (Performance difference lemma (Kakade & Langford, 2002)). *Let $\Delta(\mathcal{S})$ be the probabilistic simplex on $\mathcal{S}$. Given two policies $\pi_1$ and $\pi_2$ and arbitrary $\rho \in \Delta(\mathcal{S})$, there holds*

$$V^{\pi_1}(\rho) - V^{\pi_2}(\rho) = \frac{1}{1-\gamma}\mathbb{E}_{s \sim d_\rho^{\pi_1}}\left[\mathcal{T}^{\pi_1}V^{\pi_2}(s) - V^{\pi_2}(s)\right]$$

$$= \frac{1}{1-\gamma}\mathbb{E}_{s \sim d_\rho^{\pi_1}}\left[\sum_{a \in \mathcal{A}} \pi_1(a|s)A^{\pi_2}(s,a)\right].$$

Denoting by $b_s^\pi$ the probability on non-optimal actions

$$b_s^\pi := \pi_s\left(\mathcal{A} \setminus \mathcal{A}_s^*\right) = \sum_{a \notin \mathcal{A}_s^*} \pi(a|s),$$

the following lemma from Liu et al. (2024a) provides a lower bound of the value error based on $b_s^\pi$.

**Lemma A.5** (Lemma 2.5 (Liu et al., 2024a)). *For any policy $\pi$ and $\rho \in \Delta(\mathcal{S})$,*

$$\mathbb{E}_{s \sim \rho}\left[b_s^\pi\right] \leq \frac{V^*(\rho) - V^\pi(\rho)}{\Delta}.$$

At last we invoke two useful results from Liu et al. (2024b).

**Lemma A.6.** *For any random variable $X$ and two real-valued functions $f$, $g$, there holds*

$$\mathrm{Cov}\left(f(X), g(X)\right) = \frac{1}{2}\mathbb{E}_{X,Y}\left[\left(f(X) - f(Y)\right)\left(g(X) - g(Y)\right)\right],$$

*where $Y$ is an i.i.d. copy of $X$.*

**Lemma A.7.** *For any random variable $X$ and two monotonically increasing functions $f$ and $g$, there holds*

$$\mathrm{Cov}\left(f(X), g(X)\right) \geq 0.$$

## B    BOUNDS OF POLICY RATIO

This section provides bounds for the policy ratio

$$\frac{\pi^{k+1}(a|s)}{\pi^k(a|s)}.$$

We first provide bounds for policy ratio over non-optimal actions. Let $\mathcal{A}'_s := \arg\max_{a' \notin \mathcal{A}^*_s} A^*(s,a)$ and $\Delta_s := \min_{a \notin \mathcal{A}^*_s} |A^*(s,a)|$ for $s \in \tilde{\mathcal{S}}$. For an arbitrarily small $\varepsilon > 0$, recall that $T(\varepsilon)$ is the time such that $\left\| V^* - V^k \right\|_\infty \leq \varepsilon$ for all $k \geq T(\varepsilon)$. In such an $\varepsilon$-sub-optimal local region, we give the following bounds for non-optimal actions.

**Lemma B.1.** *Suppose the assumptions* (I), (II) *hold. For small enough $\varepsilon > 0$, when $k \geq T(\varepsilon)$ we have*

$$\forall s \in \tilde{\mathcal{S}}, \ a' \notin \mathcal{A}^*_s : \quad \frac{\pi^{k+1}(a'|s)}{\pi^k(a'|s)} \leq \frac{\phi\left(-\eta(\Delta_s - \varepsilon)\right)}{\phi\left(-\eta\varepsilon\right)} \cdot \left(1 - \frac{\varepsilon}{\Delta}\right)^{-1} := \rho_1(\eta, \Delta_s, \varepsilon) , \tag{3}$$

*and*

$$\forall s \in \tilde{\mathcal{S}}, \ a' \in \mathcal{A}'_s : \quad \frac{\pi^{k+1}(a'|s)}{\pi^k(a'|s)} \geq \frac{\phi\left(-\eta(\Delta_s + \varepsilon)\right)}{\phi\left(\eta\varepsilon\right)} := \rho_2(\eta, \Delta_s, \varepsilon) . \tag{4}$$

**Remark B.1.** *The action set $\mathcal{A}'_s$ represents "the best non-optimal actions at state $s$". Eq (3) shows that the policy on all non-optimal actions converges to zero with rate of $\rho_1$ after $k \geq T(\varepsilon)$, and Eq (4) gives the lower bound of the convergence rate of policy on $\mathcal{A}'_s$.*

**Remark B.2.** *For softmax NPG, there has been dimension-free convergence results (see Agarwal et al. (2021); Xiao (2022) for instance) due to the good property of its mirror descent form. Thus a concise upper bound of $T(\varepsilon)$ can be computed (Khodadadian et al., 2021). However, it is difficult to estimate $T(\varepsilon)$ for general $\phi$-updates, since such concise convergence result does not hold.*

*Proof.* From $\left\| V^* - V^k \right\|_\infty \leq \varepsilon$ we have $\left\| A^* - A^k \right\|_\infty \leq \varepsilon$ by Lemma A.2. Recalling the definition of $\Delta_s$, one has

$$\forall a' \notin \mathcal{A}^*_s : \quad A^k(s,a') \leq A^*(s,a') + \varepsilon \leq -\Delta_s + \varepsilon,$$
$$\forall a' \in \mathcal{A}'_s : \quad A^k(s,a') \geq A^*(s,a') - \varepsilon \geq -\Delta_s - \varepsilon,$$
$$\forall a \in \mathcal{A}^*_s : \quad -\varepsilon \leq A^k(s,a) \leq \varepsilon,$$
$$-\varepsilon \leq \max_a A^k(s,a) \leq \varepsilon.$$

Note that

$$\forall a' : \quad \frac{\pi^{k+1}(a'|s)}{\pi^k(a'|s)} = \frac{\phi\left(\eta A^k(s,a')\right)}{\mathbb{E}_{a \sim \pi^k(\cdot|s)}\left[\phi\left(\eta A^k(s,a)\right)\right]}.$$

By Lemma A.5, for any $\rho \in \Delta(\mathcal{S})$ and $s \in \mathcal{S}$,

$$\Delta \cdot \mathbb{E}_{s \sim \rho}[b^k_s] \leq \left\| V^* - V^k \right\|_\infty \leq \varepsilon.$$

Considering $\rho$ as the one-point distribution on $s$, one has

$$b^k_s \leq \frac{\varepsilon}{\Delta}. \tag{5}$$

Hence for sufficiently small $\varepsilon$,

$$\begin{aligned}
\mathbb{E}_{a \sim \pi^k(\cdot|s)}\left[\phi\left(\eta A^k(s,a)\right)\right] &\geq \sum_{a \in \mathcal{A}^*_s} \pi^k(a|s)\,\phi\left(\eta A^k(s,a)\right) \\
&\geq \phi(-\eta\varepsilon) \sum_{a \in \mathcal{A}^*_s} \pi^k(a|s) \\
&= \left(1 - b^k_s\right)\phi(-\eta\varepsilon) \\
&\geq \left(1 - \frac{\varepsilon}{\Delta}\right)\phi(-\eta\varepsilon),
\end{aligned}$$

and for any $a' \notin \mathcal{A}_s^*$

$$\phi\left(\eta_s^k A^k\left(s, a\right)\right) \leq \phi\left(-\eta\left(\Delta_s - \varepsilon\right)\right)$$

which yields the result. On the other hand, it is evident that

$$\forall a' \in \mathcal{A}_s' : \quad \frac{\pi^{k+1}(a'|s)}{\pi^k(a'|s)} \geq \frac{\phi\left(-\eta\left(\Delta_s + \varepsilon\right)\right)}{\phi\left(\eta\varepsilon\right)},$$

which completes the proof. □

Next, we derive bounds for policy ratio over optimal actions.

**Lemma B.2.** *Suppose the assumptions* (I), (II) *hold. Denote* $\varepsilon_k = \left\|V^* - V^k\right\|_\infty$. *For any* $s \in \mathcal{S}$ *and the optimal action* $a^* \in \mathcal{A}_s^*$, *when* $\varepsilon_k < \Delta$ *we have*

$$\frac{\pi^{k+1}(a^*|s)}{\pi^k(a^*|s)} \geq \frac{\phi(-\eta\varepsilon_k)}{\phi(\eta\varepsilon_k) + \dfrac{\varepsilon_k}{\Delta}\phi(-\eta(\Delta - \varepsilon_k))},$$

*and*

$$\frac{\pi^{k+1}(a^*|s)}{\pi^k(a^*|s)} \leq \frac{\phi(\eta\varepsilon_k)}{\left(1 - \dfrac{\varepsilon_k}{\Delta}\right)\phi(-\eta\varepsilon_k)}.$$

*Proof.* By the formular of $\phi$-update:

$$\frac{\pi^{k+1}(a^*|s)}{\pi^k(a^*|s)} = \frac{\phi(\eta A^k(s, a^*))}{\mathbb{E}_{a\sim\pi^k}\left[\phi(\eta A^k(s, a))\right]}$$

$$= \frac{\phi(\eta A^k(s, a^*))}{\sum_{a^*\in\mathcal{A}_s^*}\pi^k(a^*|s)\phi(\eta A^k(s, a^*)) + \sum_{a'\notin\mathcal{A}_s^*}\pi^k(a'|s)\phi(\eta A^k(s, a'))}. \tag{6}$$

We have $\left\|A^* - A^k\right\|_\infty \leq \left\|V^* - V^k\right\|_\infty = \varepsilon_k$, thus by the monotonicity of $\phi$ there holds

$$\phi(-\eta\varepsilon_k) \leq \phi(\eta A^k(s, a^*)) \leq \phi(\eta\varepsilon_k),$$

$$(1 - b_s^k)\phi(-\eta\varepsilon_k) \leq \sum_{a^*\in\mathcal{A}_s^*}\pi^k(a^*|s)\phi(\eta A^k(s, a^*)) \leq (1 - b_s^k)\phi(\eta\varepsilon_k),$$

and

$$0 \leq \sum_{a'\notin\mathcal{A}_s^*}\pi^k(a'|s)\phi(\eta A^k(s, a')) \leq b_s^k \cdot \phi(-\eta(\Delta - \varepsilon_k)).$$

Plugging them into Eq (6) and using $0 \leq b_s^k \leq \varepsilon_k/\Delta$ (Eq (5)) yield the results. □

# C  PROOFS OF RESULTS IN SECTION 3.1

## C.1  PROOF OF PROPOSITION 3.1

The first claim follows directly from (we use $A^k_{s,a}$ to represent $A^k(s,a)$ for short)

$$
\begin{aligned}
\sum_a \pi^{k+1}(a|s)A^k_{s,a} &= \frac{\mathbb{E}_{a\sim\pi^k(\cdot|s)}\left[A^k_{s,a}\phi(\eta A^k_{s,a})\right]}{\mathbb{E}_{a\sim\pi^k(\cdot|s)}\left[\phi(\eta A^k_{s,a})\right]} \\
&= \frac{\mathrm{Cov}_{a\sim\pi^k(\cdot|s)}\left(A^k_{s,a},\,\phi\left(\eta A^k_{s,a}\right)\right) + \mathbb{E}_{a\sim\pi^k(\cdot|s)}\left[A^k_{s,a}\right]\mathbb{E}_{a\sim\pi^k(\cdot|s)}\left[\phi(\eta A^k_{s,a})\right]}{\mathbb{E}_{a\sim\pi^k(\cdot|s)}\left[\phi(\eta A^k_{s,a})\right]} \\
&= \frac{\mathrm{Cov}_{a\sim\pi^k(\cdot|s)}\left(A^k_{s,a},\,\phi\left(\eta A^k_{s,a}\right)\right)}{\mathbb{E}_{a\sim\pi^k(\cdot|s)}\left[\phi(\eta A^k_{s,a})\right]} \geq 0, \qquad \text{(Lemma A.7)}
\end{aligned}
$$

where the last line leverages the monotonicity and strict positivity assumptions of $\phi$. The second claim follows directly from the performance difference lemma (Lemma A.4).

## C.2  PROOF OF THEOREM 3.1

**Additional notations**  By Proposition 3.1, we know that the value function $V^k$ is monotonically increasing. Since $V^k$ is bounded (Lemma A.1), it's easy to know that limit value function exists. We define

$$(1)\; V^\infty := \lim_{k\to+\infty} V^k, \qquad (2)\; Q^\infty := \lim_{k\to+\infty} Q^k, \qquad (3)\; A^\infty := \lim_{k\to+\infty} A^k.$$

For any state $s\in\mathcal{S}$, we follow Agarwal et al. (2021) to define the following three sets:

$$
\begin{aligned}
&(1)\; I^+_s := \left\{a\in\mathcal{A}: A^\infty(s,a) > 0\right\}, \\
&(2)\; I^0_s := \left\{a\in\mathcal{A}: A^\infty(s,a) = 0\right\}, \\
&(3)\; I^-_s := \left\{a\in\mathcal{A}: A^\infty(s,a) < 0\right\}.
\end{aligned}
$$

Finally, for some action set $\hat{\mathcal{A}} \in \left\{I^+_s, I^0_s, I^-_s\right\}$, we define

$$l^k_s\left(\hat{\mathcal{A}}\right) := \sum_{a\in\hat{\mathcal{A}}} \pi^k(a|s)\,\phi\left(\eta A^k(s,a)\right),$$

and

$$\pi^k_s\left(\hat{\mathcal{A}}\right) := \sum_{a\in\hat{\mathcal{A}}} \pi^k(a|s).$$

**Proof sketch**  Similar to Agarwal et al. (2021), the overall proof idea is to show that $\max_a A^\infty(s,a) \leq 0$ for all $s\in\mathcal{S}$, then the global convergence is given by the performance difference lemma (Lemma A.4). By the definition of $I^+_s$, it suffices to prove that $I^+_s$ is empty for any $s\in\mathcal{S}$.

- We use contradiction and assume there exists a state $s_0$ where $I^+_{s_0}$ is not empty. We prove a lemma to show that $\inf_k \pi^k_s\left(I^+_{s_0}\right) > 0$ and $\inf_k \pi^k_s\left(I^-_{s_0}\right) > 0$ (Lemma C.1).
- Noting that $V^{k+1}(\mu) - V^k(\mu) \to 0$, one can show that $\sum_a \pi^{k+1}(a|s)A^k(s,a) \to 0$ for all state $s$.
- With Lemma C.1, a direct computation shows that there exists a time $T_0$ such that $\sum_a \pi^{k+1}(a|s)A^k(s,a) > c$ with some positive constant $c$ holds for all $k \geq T_0$. It contradicts $\sum_a \pi^{k+1}(a|s)A^k(s,a) \to 0$ so the proof is completed.

In Agarwal et al. (2021), the contradiction is performed by discussing the limits of softmax parameters. Hence the proof cannot directly apply in our case, where $\phi$-update is performed in the policy space.

**Lemma C.1.** *For any $s \in \mathcal{S}$, if $I_s^+$ is non-empty, then $\inf\limits_k \pi_s^k(I_s^+) > 0$ and $\inf\limits_k \pi_s^k(I_s^-) > 0$.*

*Proof.* According to the definition of $\phi$-update, one has

$$
\begin{aligned}
\pi_s^{k+1}\left(I_s^+\right) &= \sum_{a \in I_s^+} \pi^k(a|s) \frac{\phi\left(\eta A^k(s,a)\right)}{\sum_{a' \in \mathcal{A}} \pi^k(a'|s) \phi\left(\eta A^k(s,a')\right)} \\
&= \frac{l_s^k\left(I_s^+\right)}{l_s^k\left(I_s^+\right) + l_s^k\left(I_s^0\right) + l_s^k\left(I_s^-\right)}.
\end{aligned}
\tag{7}
$$

By the definitions of $I_s^+$, $I_s^0$ and $I_s^-$, it is obvious that there exists a time $T_0 \in \mathbb{N}$ and a constant $\varepsilon > 0$ such that for all $k \geq T_0$ ($A_{s,a}^k$ is short for $A^k(s,a)$):

$$
(1)\, \forall\, a \in I_s^+ : A_{s,a}^k > \varepsilon, \qquad (2)\, \forall\, a \in I_s^0 : \left|A_{s,a}^k\right| < \frac{\varepsilon}{2}, \qquad (3)\, \forall\, a \in I_s^- : A_{s,a}^k < -\varepsilon.
\tag{8}
$$

By the monotonicity assumption of $\phi$ and equation (8),

$$
\begin{aligned}
l_s^k\left(I_s^+\right) &\geq \pi_s^k\left(I_s^+\right) \cdot \phi(\eta\varepsilon), \\
l_s^k\left(I_s^0\right) &\leq \pi_s^k\left(I_s^0\right) \cdot \phi\left(\eta\frac{\varepsilon}{2}\right) < \pi_s^k\left(I_s^0\right) \cdot \phi(\eta\varepsilon), \\
l_s^k\left(I_s^-\right) &\leq \pi_s^k\left(I_s^-\right) \cdot \phi(-\eta\varepsilon) < \pi_s^k\left(I_s^-\right) \cdot \phi(\eta\varepsilon).
\end{aligned}
$$

Plugging it into (7) yields that

$$
\begin{aligned}
\pi_s^{k+1}\left(I_s^+\right) &\geq \frac{\pi_s^k\left(I_s^+\right) \cdot \phi(\eta\varepsilon)}{\pi_s^k\left(I_s^+\right) \cdot \phi(\eta\varepsilon) + l_s^k\left(I_s^0\right) + l_s^k\left(I_s^-\right)} \\
&\geq \frac{\pi_s^k\left(I_s^+\right) \cdot \phi(\eta\varepsilon)}{\left(\pi_s^k\left(I_s^+\right) + \pi_s^k\left(I_s^0\right) + \pi_s^k\left(I_s^-\right)\right) \cdot \phi(\eta\varepsilon)} \\
&= \pi_s^k\left(I_s^+\right).
\end{aligned}
$$

Thus the sequence $\left\{\pi_s^k\left(I_s^+\right) : k \geq T_0\right\}$ is monotonically increasing, and one has

$$
\inf_k \pi_s^k\left(I_s^+\right) = \min_{k \leq T_0} \pi_s^k\left(I_s^+\right) > 0
$$

as $\pi^k(a|s) > 0$ for any finite $k$. It's obvious that $\inf\limits_k \pi_s^k(I_s^-) > 0$. If not, as $\pi^k(a|s) > 0$, one has $\liminf\limits_{k \to +\infty} \pi_s^k(I_s^-) = 0$ and then

$$
\begin{aligned}
\liminf_{k \to +\infty}\left\{\sum_a \pi^k A^k(s,a)\right\} &= \liminf_{k \to +\infty}\left\{\sum_{a \in I_s^+} \pi^k A^k(s,a) + \sum_{a \in I_s^0} \pi^k A^k(s,a) + \sum_{a \in I_s^-} \pi^k A^k(s,a)\right\} \\
&= \liminf_{k \to +\infty}\left\{\sum_{a \in I_s^+} \pi^k A^k(s,a)\right\} \\
&> 0
\end{aligned}
$$

which contradicts $\forall k \in \mathbb{N} : \sum_a \pi^k(a|s) A^k(s,a) = 0$. $\qquad \square$

*Proof of Theorem 3.1.* As $V^k$ is monotonically increasing and $V^\infty = \lim_{k\to\infty} V^k$ exists, we have

$$\lim_{k\to\infty} V^{k+1}(\mu) - V^k(\mu) = 0. \tag{9}$$

By the performance difference lemma (Lemma A.4),

$$V^{k+1}(\mu) - V^k(\mu) = \frac{1}{1-\gamma} \sum_s d_\mu^k(s) \sum_a \pi^{k+1}(a|s) A^k(s,a).$$

As $d_\mu^k(s) > 0$ and $\sum_a \pi^{k+1}(a|s) A^k(s,a) \geq 0$, it can be verified that equation (9) implies

$$\forall s \in \mathcal{S}: \quad \sum_a \pi^{k+1}(a|s) A^k(s,a) \to 0. \tag{10}$$

In the following, we will show that if there exists a state $s_0 \in \mathcal{S}$ such that $I_{s_0}^+$ is non-empty, then the condition equation (10) is violated, hence there must be $I_s^+ = \varnothing$ for all $s$.

By Lemma C.1, we know that

$$c_1 := \inf_k \pi_{s_0}^k \left( I_{s_0}^+ \right) > 0 \qquad \text{and} \qquad c_2 := \inf_k \pi_{s_0}^k \left( I_{s_0}^- \right) > 0,$$

Recalling the definition of $T_0$, $\varepsilon$ and $l_{s_0}^k$, for all $k \geq T_0$,

$$\sum_{a \in \mathcal{A}} \pi^{k+1}(a|s_0) A^k(s_0,a) = \frac{1}{Z_{s_0}^k} \mathbb{E}_{a\sim\pi^k(\cdot|s_0)} \left[ \phi\left( \eta A^k(s_0,a) \right) A^k(s_0,a) \right]$$

$$= \frac{1}{Z_{s_0}^k} \mathrm{Cov}_{a\sim\pi^k(\cdot|s_0)} \left( \phi\left( \eta A^k(s_0,a) \right), A^k(s_0,a) \right).$$

Let $a'$ be an i.i.d. copy of $a$. Then by Lemma A.6,

$$\sum_{a \in \mathcal{A}} \pi^{k+1}(a|s_0) A^k(s_0,a)$$

$$= \frac{1}{2Z_{s_0}^k} \mathbb{E}_{a\sim\pi^k(\cdot|s_0),a'\sim\pi^k(\cdot|s_0)} \left[ \left( A^k(s_0,a) - A^k(s_0,a') \right) \left( \phi\left( \eta A^k(s_0,a) \right) - \phi\left( \eta A^k(s_0,a') \right) \right) \right]$$

$$\geq \frac{1}{2Z_{s_0}^k} \sum_{a \in I_{s_0}^+} \sum_{a' \in I_{s_0}^-} \pi^k(a|s_0) \pi^k(a'|s_0) \left[ \left( A^k(s_0,a) - A^k(s_0,a') \right) \left( \phi\left( \eta A^k(s_0,a) \right) - \phi\left( \eta A^k(s_0,a') \right) \right) \right]$$

$$\geq \frac{1}{2Z_{s_0}^k} \sum_{a \in I_{s_0}^+} \sum_{a' \in I_{s_0}^-} \pi^k(a|s_0) \pi^k(a'|s_0) \left[ \left( \varepsilon - (-\varepsilon) \right) \left( \phi\left( \eta A^k(s_0,a) \right) - \phi(-\eta\varepsilon) \right) \right]$$

$$= \frac{\varepsilon}{Z_{s_0}^k} \sum_{a \in I_{s_0}^+} \sum_{a' \in I_{s_0}^-} \pi^k(a|s_0) \pi^k(a'|s_0) \left( \phi\left( \eta A^k(s_0,a) \right) - \phi(-\eta\varepsilon) \right)$$

$$= \frac{\varepsilon}{Z_{s_0}^k} \sum_{a \in I_{s_0}^+} \pi^k(a|s_0) \left( \phi\left( \eta A^k(s_0,a) \right) - \phi(-\eta\varepsilon) \right) \pi_{s_0}^k \left( I_{s_0}^- \right)$$

$$\geq \frac{c_2\varepsilon}{Z_{s_0}^k} \sum_{a \in I_{s_0}^+} \pi^k(a|s_0) \left( \phi\left( \eta A^k(s_0,a) \right) - \phi(-\eta\varepsilon) \right)$$

$$= \frac{c_2\varepsilon}{Z_{s_0}^k} \left( l_{s_0}^k \left( I_{s_0}^+ \right) - \pi_{s_0}^k \left( I_{s_0}^+ \right) \phi(-\eta\varepsilon) \right).$$

In the derivation above $\left( A^k \left( s_0, a \right) - A^k \left( s_0, a' \right) \right) \left( \phi \left( \eta A^k \left( s_0, a \right) \right) - \phi \left( \eta A^k \left( s_0, a' \right) \right) \right) \geq 0$ for any $a, a'$ by the monotonicity of $\phi$. Now notice that

$$Z^k_{s_0} = l^k_{s_0} \left( I^+_{s_0} \right) + l^k_{s_0} \left( I^0_{s_0} \right) + l^k_{s_0} \left( I^-_{s_0} \right).$$

Thus for $k \geq T_0$,

$$\sum_{a \in \mathcal{A}} \pi^{k+1}(a|s_0) A^k \left( s_0, a \right) \geq \varepsilon \cdot c_2 \cdot \frac{1 - \dfrac{\pi^k_{s_0} \left( I^+_{s_0} \right) \phi \left( -\eta \varepsilon \right)}{l^k_s \left( I^+_{s_0} \right)}}{1 + \dfrac{l^k_s \left( I^0_{s_0} \right)}{l^k_s \left( I^+_{s_0} \right)} + \dfrac{l^k_s \left( I^-_{s_0} \right)}{l^k_s \left( I^+_{s_0} \right)}}$$

$$\geq \varepsilon \cdot c_2 \cdot \frac{1 - \dfrac{\pi^k_{s_0} \left( I^+_{s_0} \right) \phi \left( -\eta \varepsilon \right)}{\pi^k_{s_0} \left( I^+_{s_0} \right) \phi \left( \eta \varepsilon \right)}}{1 + \dfrac{l^k_s \left( I^0_{s_0} \right)}{l^k_s \left( I^+_{s_0} \right)} + \dfrac{l^k_s \left( I^-_{s_0} \right)}{l^k_s \left( I^+_{s_0} \right)}}$$

$$\geq \varepsilon \cdot c_2 \cdot \frac{1 - \dfrac{\phi \left( -\eta \varepsilon \right)}{\phi \left( \eta \varepsilon \right)}}{1 + \dfrac{\pi^k_{s_0} \left( I^0_{s_0} \right) \phi \left( \eta^k_s \varepsilon \right)}{\pi^k_{s_0} \left( I^+_{s_0} \right) \phi \left( \eta^k_s \varepsilon \right)} + \dfrac{\pi^k_{s_0} \left( I^-_{s_0} \right) \phi \left( \eta^k_s \varepsilon \right)}{\pi^k_{s_0} \left( I^+_{s_0} \right) \phi \left( \eta^k_s \varepsilon \right)}}$$

$$= \varepsilon \cdot c_2 \cdot \frac{1 - \dfrac{\phi \left( -\eta \varepsilon \right)}{\phi \left( \eta \varepsilon \right)}}{1 + \dfrac{\pi^k_{s_0} \left( I^0_{s_0} \right)}{\pi^k_{s_0} \left( I^+_{s_0} \right)} + \dfrac{\pi^k_{s_0} \left( I^-_{s_0} \right)}{\pi^k_{s_0} \left( I^+_{s_0} \right)}}$$

$$= \varepsilon \cdot c_2 \cdot \pi^k_{s_0} \left( I^+_{s_0} \right) \cdot \left( 1 - \frac{\phi \left( -\eta \varepsilon \right)}{\phi \left( \eta \varepsilon \right)} \right)$$

$$\geq \varepsilon \cdot c_1 \cdot c_2 \cdot \left( 1 - \frac{\phi \left( -\eta \varepsilon \right)}{\phi \left( \eta \varepsilon \right)} \right) > 0,$$

which contradicts equation (10). Thus for all $s \in \mathcal{S}$, one has $\max\limits_{s,a} A^\infty \left( s, a \right) \leq 0$ and the global convergence can be obtained by Lemma A.4 easily. □

# D  PROOFS OF RESULTS IN SECTION 3.2

## D.1  PROOF OF THEOREM 3.2

Combining Theorem 3.1 and Lemma B.1 we know that there exists a time $T(\varepsilon) \in \mathbb{N}$ such that

$$\forall\, k \geq T \left( \varepsilon \right), \; s \in \tilde{\mathcal{S}}, \; a \notin \mathcal{A}^*_s : \quad \pi^k \left( a|s \right) \leq \rho_1(\eta, \Delta_s, \varepsilon) \pi^{k-1} \left( a|s \right)$$
$$\leq \rho_1(\eta, \Delta, \varepsilon) \pi^{k-1} \left( a|s \right)$$
$$\leq \ldots$$
$$\leq \rho_1 \left( \eta, \Delta, \varepsilon \right)^{(k-T(\varepsilon))} \cdot \pi^{T(\varepsilon)} \left( a|s \right).$$

By Lemma A.4, for all $k \geq T(\varepsilon)$,

$$
\begin{aligned}
V^* (\mu) - V^k (\mu) &\leq \frac{1}{1-\gamma} \sum_{s \in \mathcal{S}} d_\mu^k (s) \sum_a \pi^k (a|s) |A^* (s,a)| \\
&= \frac{1}{1-\gamma} \sum_{s \in \tilde{\mathcal{S}}} d_\mu^k(s) \sum_{a \notin \mathcal{A}_s^*} \pi^k(a|s) |A^*(s,a)| \qquad (A^*(s,a) = 0 \text{ for } s \notin \tilde{\mathcal{S}}) \\
&\leq \frac{\rho_1 (\eta, \Delta, \varepsilon)^{(k-T(\varepsilon))}}{1-\gamma} \cdot \sum_{s \in \tilde{\mathcal{S}}} d_\mu^k (s) \sum_{a \notin \mathcal{A}_s^*} \pi^{T(\varepsilon)} (a|s) |A^* (s,a)| \\
&= \frac{\rho_1 (\eta, \Delta, \varepsilon)^{(k-T(\varepsilon))}}{1-\gamma} \cdot \sum_{s \in \tilde{\mathcal{S}}} \frac{d_\mu^k (s)}{d_\mu^{T(\varepsilon)} (s)} d_\mu^{T(\varepsilon)} (s) \sum_{a \notin \mathcal{A}_s^*} \pi^{T(\varepsilon)} (a|s) |A^* (s,a)| \\
&\leq \frac{\rho_1 (\eta, \Delta, \varepsilon)^{(k-T(\varepsilon))}}{1-\gamma} \cdot \sum_{s \in \tilde{\mathcal{S}}} \frac{1}{(1-\gamma) \tilde{\mu}} \cdot d_\mu^{T(\varepsilon)} (s) \sum_{a \notin \mathcal{A}_s^*} \pi^{T(\varepsilon)} (a|s) |A^* (s,a)| \\
&= \frac{\rho_1 (\eta, \Delta, \varepsilon)^{(k-T(\varepsilon))}}{(1-\gamma) \tilde{\mu}} \cdot \left( V^* (\mu) - V^{T(\varepsilon)} (\mu) \right). \\
&\leq \frac{\rho_1 (\eta, \Delta, \varepsilon)^{(k-T(\varepsilon))}}{(1-\gamma) \tilde{\mu}} \cdot \left\| V^* - V^{T(\varepsilon)} \right\|_\infty.
\end{aligned}
$$

By Lemma A.3,

$$
\begin{aligned}
\forall k \geq T(\varepsilon): \quad \left\| V^* - V^k \right\|_\infty &\leq \frac{1}{\tilde{\mu}} \left( V^* (\mu) - V^k (\mu) \right) \\
&\leq \frac{\rho_1 (\eta, \Delta, \varepsilon)^{(k-T(\varepsilon))}}{(1-\gamma) \tilde{\mu}^2} \cdot \left\| V^* - V^{T(\varepsilon)} \right\|_\infty \qquad (11) \\
&\leq \frac{\rho_1 (\eta, \Delta, \varepsilon)^{(k-T(\varepsilon))}}{(1-\gamma)^2 \tilde{\mu}^2}.
\end{aligned}
$$

## D.2 PROOF OF THEOREM 3.3

**Proof sketch** By the global asymptotic convergence result (Theorem 3.1), the policy value of sub-optimal actions vanish, i.e. $\pi^k(a'|s) \to 0$ for all $s$ and $a' \notin \mathcal{A}_s^*$. Thus it suffices to show that $\pi^k(a^*|s)$ converges for any state $s \in \mathcal{S}$ and optimal action $a^* \in \mathcal{A}_s^*$. Denote $\varepsilon_k := \left\| V^* - V^k \right\|_\infty$.

- Recall that Lemma B.2 characterizes the contraction rate of $\pi^k(a^*|s)$ in the local region (i.e. $\varepsilon_k$ is small). It is shown that

$$
\frac{\phi(-\eta \varepsilon_k)}{\phi(\eta \varepsilon_k)} \lesssim \frac{\pi^{k+1}(a^*|s)}{\pi^k(a^*|s)} \lesssim \frac{\phi(\eta \varepsilon_k)}{\phi(-\eta \varepsilon_k)}.
$$

- The crucial part is to show that $\left\{ \log \pi^k(a^*|s) \right\}$ is a Cauchy sequence. Fix a sufficiently small $\varepsilon$. By Lemma B.2 and the assumption (III) for $\phi$, it can be shown that

$$
\left| \log \pi^{T(\varepsilon)+t+1}(a^*|s) - \log \pi^{T(\varepsilon)+t}(a^*|s) \right| \lesssim \varepsilon_{T(\varepsilon)+t}.
$$

By the local linear convergence we have $\varepsilon_{T(\varepsilon)+t} \lesssim \rho_1^t \cdot \varepsilon$. Basing it we show that

$$\sum_{k=T(\varepsilon)}^{\infty} \left| \log \pi^{k+1}(a^*|s) - \log \pi^l(a^*|s) \right| \lesssim \varepsilon$$

which verifies the Cauchy property.

- As $\log \pi^k(a^*|s)$ converges, we know that $\log \pi^k(a^*|s) > -\infty$, thus $\pi^k(a^*|s)$ also converges. For the sub-optimal actions $a' \notin \mathcal{A}_s^*$, there holds $\pi^k(a'|s) \to 0$ indeed. Hence the policy convergence is established.

*Proof.* As we have established the global asymptotic convergence (i.e. $V^k \to V^*$, Theorem 3.1), we can conclude that $\pi^k(a'|s) \to 0$ for all $s \in \mathcal{S}$ and $a' \notin \mathcal{A}_s^*$. Thus it suffices to show that $\pi^k(a^*|s)$ converges for all $s \in \mathcal{S}$ and optimal actions $a^* \in \mathcal{A}_s^*$.

First note that $\pi^k(a|s) > 0$ for all $k$. Thus, $\log \pi^k(a|s)$ is properly defined. Additionally, it is not hard to see that $\rho_1(\eta, \Delta, \varepsilon)$ decreases as $\varepsilon$ decreases, thus there exists a $\varepsilon_0$ such that for any $0 < \varepsilon \leq \varepsilon_0$ we have $\rho_1(\eta, \Delta, \varepsilon) \leq \rho_1(\eta, \Delta, \varepsilon_0) < 1$. Now we are going to show that $\{\log \pi^k(a^*|s)\}$ is a Cauchy sequence. For arbitrary $\varepsilon > 0$ which is small enough to satisfy $\varepsilon \leq \min\{\delta/\eta, \ \Delta/2, \ \varepsilon_0\}$, there exists a time $T(\varepsilon)$ such that

$$\forall\, k \geq T(\varepsilon): \quad \varepsilon_k := \left\| V^* - V^k \right\|_\infty \leq \varepsilon.$$

Furthermore, by the local convergence rate result (Eq (11)) we have

$$\forall\, k \geq T(\varepsilon): \quad \varepsilon_k \leq \frac{\varepsilon}{(1-\gamma)\tilde{\mu}^2}\rho_1(\eta, \Delta, \varepsilon)^{k-T(\varepsilon)}. \tag{12}$$

Now we consider the summation

$$\sum_{k=T(\varepsilon)}^{\infty} \left| \log \pi^{k+1}(a^*|s) - \log \pi^k(a^*|s) \right|.$$

By Lemma B.2, for any $k \geq T(\varepsilon)$ one has

$$\left| \log \pi^{k+1}(a^*|s) - \log \pi^k(a^*|s) \right|$$
$$\leq \log \max \left\{ \frac{\phi(\eta\varepsilon_k) + \dfrac{\varepsilon_k}{\Delta}\phi(-\eta(\Delta - \varepsilon_k))}{\phi(-\eta\varepsilon_k)}, \ \frac{\phi(\eta\varepsilon_k)}{\left(1 - \dfrac{\varepsilon_k}{\Delta}\right)\phi(-\eta\varepsilon_k)} \right\}$$
$$\leq \log \frac{\phi(\eta\varepsilon_k) + \dfrac{\varepsilon_k}{\Delta}\phi(-\eta(\Delta - \varepsilon_k))}{\phi(-\eta\varepsilon_k)} + \log \frac{\phi(\eta\varepsilon_k)}{\left(1 - \dfrac{\varepsilon_k}{\Delta}\right)\phi(-\eta\varepsilon_k)}.$$

For the first term, noting $\varepsilon_k \leq \varepsilon \leq \Delta/2$ one has

$$\sum_{k=T(\varepsilon)}^{\infty} \log \frac{\phi(\eta \varepsilon_k) + \dfrac{\varepsilon_k}{\Delta} \phi(-\eta(\Delta - \varepsilon_k))}{\phi(-\eta \varepsilon_k)} \leq \sum_{k=T(\varepsilon)}^{\infty} \log \left[ \frac{\phi(\eta \varepsilon_k)}{\phi(-\eta \varepsilon_k)} + \frac{\varepsilon_k}{\Delta} \right]$$

$$\leq \sum_{k=T(\varepsilon)}^{\infty} \log \left[ \left( 1 + \frac{\varepsilon_k}{\Delta} \right) \frac{\phi(\eta \varepsilon_k)}{\phi(-\eta \varepsilon_k)} \right]$$

$$= \sum_{k=T(\varepsilon)}^{\infty} \log \left[ \left( 1 + \frac{\varepsilon_k}{\Delta} \right) \right] + \sum_{k=T(\varepsilon)}^{\infty} \log \left[ \frac{\phi(\eta \varepsilon_k)}{\phi(-\eta \varepsilon_k)} \right]$$

$$\leq \frac{1}{\Delta} \sum_{k=T(\varepsilon)}^{\infty} \varepsilon_k + \sum_{k=T(\varepsilon)}^{\infty} \log \frac{\phi(\eta \varepsilon_k)}{\phi(-\eta \varepsilon_k)}.$$

By the assumption (III) and $\varepsilon \leq \delta/\eta$, we have

$$\log \frac{\phi(\eta \varepsilon_k)}{\phi(-\eta \varepsilon_k)} = \log \phi(\eta \varepsilon_k) - \log \phi(-\eta \varepsilon_k) = (2\eta \varepsilon_k) \cdot \frac{d}{dt} \log \phi(\xi_k) \leq 2c\eta \varepsilon_k \qquad (13)$$

where $\xi_k \in [-\eta \varepsilon_k, \eta \varepsilon_k]$. Using Eq (10) and Eq (12) we get (noting that $\phi(\eta, \Delta, \varepsilon) < 1$)

$$\sum_{k=T(\varepsilon)}^{\infty} \log \frac{\phi(\eta \varepsilon_k) + \dfrac{\varepsilon_k}{\Delta} \phi(-\eta(\Delta - \varepsilon_k))}{\phi(-\eta \varepsilon_k)} \leq \left( \frac{1}{\Delta} + 2c\eta \right) \sum_{k=T(\varepsilon)}^{\infty} \varepsilon_k$$

$$\leq \frac{\frac{1}{\Delta} + 2c\eta}{(1-\gamma)\tilde{\mu}^2} \frac{\varepsilon}{1 - \rho_1(\eta, \Delta, \varepsilon)}.$$

For the second term, one has

$$\sum_{k=T(\varepsilon)}^{\infty} \log \frac{\phi(\eta \varepsilon_k)}{\left( 1 - \dfrac{\varepsilon_k}{\Delta} \right) \phi(-\eta \varepsilon_k)} = \sum_{k=T(\varepsilon)}^{\infty} \log \frac{\phi(\eta \varepsilon_k)}{\phi(-\eta \varepsilon_k)} - \sum_{k=T(\varepsilon)}^{\infty} \log \left( 1 - \frac{\varepsilon_k}{\Delta} \right)$$

$$\leq \sum_{k=T(\varepsilon)}^{\infty} \log \frac{\phi(\eta \varepsilon_k)}{\phi(-\eta \varepsilon_k)} + \frac{1}{\Delta} \sum_{k=T(\varepsilon)}^{\infty} \varepsilon_k$$

$$\leq \frac{\frac{1}{\Delta} + 2c\eta}{(1-\gamma)\tilde{\mu}^2} \frac{\varepsilon}{1 - \rho_1(\eta, \Delta, \varepsilon)},$$

where the first inequality uses $\varepsilon_k \leq \Delta/2$ and the fact that

$$\forall\, 0 \leq x \leq 1/2: \quad -\log(1-x) \leq x.$$

Putting them together yields

$$\sum_{k=T(\varepsilon)}^{\infty} \left| \log \pi^{k+1}(a^*|s) - \log \pi^k(a^*|s) \right| \leq \frac{\frac{1}{\Delta} + 2c\eta}{(1-\gamma)\tilde{\mu}^2} \frac{2\varepsilon}{1 - \rho_1(\eta, \Delta, \varepsilon)}$$

$$\leq \frac{\frac{1}{\Delta} + 2c\eta}{(1-\gamma)\tilde{\mu}^2} \frac{2\varepsilon}{1 - \rho_1(\eta, \Delta, \varepsilon_0)} := C_0 \varepsilon.$$

Therefore, after a variable substitution, it has shown that $\{\log \pi^k(a^*|s)\}$ is a Cauchy sequence. Hence $\pi^k(a^*|s)$ converges for any $s \in \mathcal{S}$ and $a^* \in \mathcal{A}_s^*$. $\qquad \square$

### D.3 PROOF OF THEOREM 3.4

**Proof sketch** We can assume that $\tilde{\mathcal{S}} = \mathcal{S}$ without loss of generality. Recall the state set $\mathcal{S}_0 = \arg\min_s \Delta_s$ and action set $\mathcal{A}'_s = \arg\max_{a' \notin \mathcal{A}^*_s} A^*(s, a')$.

- We first prove Lemma D.1, which tells that when $k \to +\infty$, the policy probability on the non-optimal actions is concentrated on $\pi^k(a_0|s_0)$ where $s_0 \in \mathcal{S}_0$ and $a_0 \in \mathcal{A}'_{s_0}$.

- Then we directly expand $V^*(\mu) - V^{k+1}(\mu)$ and $V^*(\mu) - V^k(\mu)$ to the weighted summation of $|A^*(s, a)|$ by the performance difference lemma (Lemma A.4).

- We consider the case that $d^k_\mu$, $A^k$ and $\pi^k$ are all in local regions, which can be acheived by large enough $k$. By direct calculation, we show that the convergence rate is bounded by $\zeta_1$ and $\zeta_2$. As $k \to +\infty$, both $\zeta_1$ and $\zeta_2$ converge to $\phi(-\eta\Delta)/\phi(0)$, so as well as the local convergence rate.

**Lemma D.1.** *Consider the $\phi$-update with constant step size $\eta^k_s = \eta > 0$. For any $s \in \mathcal{S}, a \in \mathcal{A}$ such that $|A^*(s, a)| > \Delta$, one has*

$$\lim_{k \to \infty} \frac{\pi^k(a|s)}{\sum_{s_0 \in \mathcal{S}_0} \sum_{a_0 \in \mathcal{A}'_{s_0}} \pi^k(a_0|s_0)} = 0.$$

*Proof.* Using the same procedure as in the proof of Lemma B.1, one can easily show that for sufficiently small $c > 0$, there exists a time $T(c)$ such that for all $k \geq T(c)$:

$$\forall s \in \mathcal{S}, \ a \notin \mathcal{A}^*_s : \quad \left| \left( \frac{\pi^{k+1}(a|s)}{\pi^k(a|s)} \middle/ \frac{\phi(-\eta|A^*(s,a)|)}{\phi(0)} \right) - 1 \right| \leq c.$$

Then for all $k \geq T(c)$,

$$\frac{\pi^k(a|s)}{\sum_{s_0 \in \mathcal{S}_0, a_0 \in \mathcal{A}'_{s_0}} \pi^k(a_0|s_0)} \leq \frac{\pi^{T(c)}(a_0|s_0) \cdot \left( \frac{\phi(-\eta|A^*(s,a)|)}{\phi(0)}(1+c) \right)^{k-T(c)}}{\sum_{s_0 \in \mathcal{S}_0, a_0 \in \mathcal{A}'_{s_0}} \pi^{T(c)}(a_0|s_0) \cdot \left( \frac{\phi(-\eta\Delta)}{\phi(0)}(1-c) \right)^{k-T(c)}}$$

$$= \frac{\pi^{T(c)}(a_0|s_0)}{\sum_{s_0 \in \mathcal{S}_0, a_0 \in \mathcal{A}'_{s_0}} \pi^{T(c)}(a_0|s_0)} \left( \frac{\frac{\phi(-\eta|A^*(s,a)|)}{\phi(0)}}{\frac{\phi(-\eta\Delta)}{\phi(0)}} \frac{1+c}{1-c} \right)^{k-T(c)}$$

$$= \frac{\pi^{T(c)}(a_0|s_0)}{\sum_{s_0 \in \mathcal{S}_0, a_0 \in \mathcal{A}'_{s_0}} \pi^{T(c)}(a_0|s_0)} \left( \frac{\phi(-\eta|A^*(s,a)|)}{\phi(-\eta\Delta)} \frac{1+c}{1-c} \right)^{k-T(c)}.$$

For any $s \in \mathcal{S}$ and $a \in \mathcal{A}$ such that $|A^*(s, a)| > \Delta$, one has $\phi(-\eta|A^*(s,a)|)/\phi(-\eta\Delta) < 1$. Thus it's trivial that

$$\lim_{k \to +\infty} \frac{\pi^k(a|s)}{\sum_{s_0 \in \mathcal{S}_0, a_0 \in \mathcal{A}'_{s_0}} \pi^k(a_0|s_0)} = 0$$

by sufficiently small $c$. □

*Proof of Theorem 3.4.* By Theorem 3.3, we have $\pi^k \to \pi^*$ for some optimal policy $\pi^*$, therefore $d^k_\mu \to d^*_\mu$. Now, for arbitrary $\varepsilon > 0, \delta > 0, \sigma > 0$, there exists a time $T(\varepsilon, \delta, \sigma) > 0$ such that

$$\forall s \in \mathcal{S}, \ k \geq T(\varepsilon, \delta, \sigma) : \quad \left| \frac{d^{k+1}_\mu(s)}{d^k_\mu(s)} - 1 \right| \leq \delta, \quad \left| \frac{d^k_\mu(s)}{d^{k+1}_\mu(s)} - 1 \right| \leq \delta,$$

$$\left\| A^* - A^k \right\|_\infty \leq \varepsilon, \quad \frac{\pi^k(a|s)}{C_k} \leq \sigma,$$

where $(s, a) \in \{(s, a) : |A^*(s, a)| > \Delta\}$. By the performance difference lemma (Lemma A.4),

$$\frac{V^*(\mu) - V^{k+1}(\mu)}{V^*(\mu) - V^k(\mu)} = \frac{\sum_s d^{k+1}_\mu(s) \sum_{a' \notin \mathcal{A}^*_s} \pi^{k+1}(a'|s) |A^*(s, a')|}{\sum_s d^k_\mu(s) \sum_{a' \notin \mathcal{A}^*_s} \pi^k(a'|s) |A^*(s, a')|}$$

$$= \frac{\sum_s d^{k+1}_\mu(s) \sum_{a' \notin \mathcal{A}^*_s} \frac{\pi^{k+1}(a'|s)}{C_k} |A^*(s, a')|}{\sum_s d^k_\mu(s) \sum_{a' \notin \mathcal{A}^*_s} \frac{\pi^k(a'|s)}{C_k} |A^*(s, a')|}.$$

By Lemma B.1,

$$\frac{V^*(\mu) - V^{k+1}(\mu)}{V^*(\mu) - V^k(\mu)}$$

$$\leq \left( 1 - \frac{\varepsilon}{\Delta \tilde{\mu}} \right)^{-1} \cdot \frac{\sum_s d^{k+1}_\mu(s) \sum_{a' \notin \mathcal{A}^*_s} \frac{\pi^k(a'|s)}{C_k} |A^*(s, a')| \left( \frac{\phi(\eta(-\Delta_s + \varepsilon))}{\phi(-\eta \varepsilon)} \right)}{\sum_s d^k_\mu(s) \sum_{a' \notin \mathcal{A}^*_s} \frac{\pi^k(a'|s)}{C_k} |A^*(s, a')|}.$$

By expanding the summation on numerator and dropping out some summation terms in denominator we get

$$\frac{V^*(\mu) - V^{k+1}(\mu)}{V^*(\mu) - V^k(\mu)}$$

$$\leq \left( 1 - \frac{\varepsilon}{\Delta \tilde{\mu}} \right)^{-1} \cdot \frac{\sum_{s \notin \mathcal{S}_0} d^{k+1}_\mu(s) \sum_{a' \notin \mathcal{A}^*_s} \frac{\pi^k(a'|s)}{C_k} |A^*(s, a')| \left( \frac{\phi(\eta(-\Delta_s + \varepsilon))}{\phi(-\eta \varepsilon)} \right)}{\sum_{s_0 \in \mathcal{S}_0} d^k_\mu(s_0) \sum_{a_0 \in \mathcal{A}'_{s_0}} \frac{\pi^k(a_0|s_0)}{C_k} |A^*(s_0, a_0)|}$$

$$+ \left( 1 - \frac{\varepsilon}{\Delta \tilde{\mu}} \right)^{-1} \cdot \frac{\sum_{s_0 \in \mathcal{S}_0} d^{k+1}_\mu(s_0) \sum_{a' \notin (\mathcal{A}^*_{s_0} \cup \mathcal{A}'_{s_0})} \frac{\pi^k(a'|s)}{C_k} |A^*(s, a')| \left( \frac{\phi(\eta(-\Delta + \varepsilon))}{\phi(-\eta \varepsilon)} \right)}{\sum_{s_0 \in \mathcal{S}_0} d^k_\mu(s_0) \sum_{a_0 \in \mathcal{A}'_{s_0}} \frac{\pi^k(a_0|s_0)}{C_k} |A^*(s_0, a_0)|}$$

$$+ \left( 1 - \frac{\varepsilon}{\Delta \tilde{\mu}} \right)^{-1} \left( \frac{\phi(\eta(-\Delta + \varepsilon))}{\phi(-\eta \varepsilon)} \right) \cdot \frac{\sum_{s_0 \in \mathcal{S}_0} d^{k+1}_\mu(s_0) \sum_{a_0 \in \mathcal{A}'_{s_0}} \pi^k(a_0|s_0) \cdot |A^*(s_0, a_0)|}{\sum_{s_0 \in \mathcal{S}_0} d^k_\mu(s_0) \sum_{a_0 \in \mathcal{A}'_{s_0}} \pi^k(a_0|s_0) \cdot |A^*(s_0, a_0)|}.$$

Noting that $|A^*(s_0, a_0)| = \Delta$, $d_\mu^{k+1}(s_0) \le d_\mu^k(s_0) + \delta$, $\pi^k(a'|s)/C_k \le \sigma$, and $|A^*(s, a')| \le 1/(1-\gamma)$, it follows that:

$$\frac{V^*(\mu) - V^{k+1}(\mu)}{V^*(\mu) - V^k(\mu)}$$

$$\le \left(1 - \frac{\varepsilon}{\Delta\tilde{\mu}}\right)^{-1} \cdot \frac{\sum_{s \notin \mathcal{S}_0} d_\mu^{k+1}(s) \sum_{a' \notin \mathcal{A}_s^*} \frac{\sigma}{1-\gamma} \cdot \frac{\phi(\eta(-\Delta_s + \varepsilon))}{\phi(-\eta\varepsilon)}}{\sum_{s_0 \in \mathcal{S}_0} d_\mu^k(s_0) \sum_{a_0 \in \mathcal{A}'_{s_0}} \frac{\pi^k(a_0|s_0)}{C_k} \Delta}$$

$$+ \left(1 - \frac{\varepsilon}{\Delta\tilde{\mu}}\right)^{-1} \cdot \frac{\sum_{s_0 \in \mathcal{S}_0} d_\mu^{k+1}(s_0) \sum_{a' \notin \left(\mathcal{A}_{s_0}^* \cup \mathcal{A}'_{s_0}\right)} \frac{\sigma}{1-\gamma} \cdot \frac{\phi(\eta(-\Delta + \varepsilon))}{\phi(-\eta\varepsilon)}}{\sum_{s_0 \in \mathcal{S}_0} d_\mu^k(s_0) \sum_{a_0 \in \mathcal{A}'_{s_0}} \frac{\pi^k(a_0|s_0)}{C_k} \Delta}$$

$$+ \left(1 - \frac{\varepsilon}{\Delta\tilde{\mu}}\right)^{-1} \left(\frac{\phi(\eta(-\Delta + \varepsilon))}{\phi(-\eta\varepsilon)}\right) \cdot \frac{\sum_{s_0 \in \mathcal{S}_0} \left(d_\mu^k(s_0) + \delta\right) \sum_{a_0 \in \mathcal{A}'_{s_0}} \pi^k(a_0|s_0) \cdot \Delta}{\sum_{s_0 \in \mathcal{S}_0} d_\mu^k(s_0) \sum_{a_0 \in \mathcal{A}'_{s_0}} \pi^k(a_0|s_0) \cdot \Delta}.$$

Noting that

$$\sum_{s_0 \in \mathcal{S}_0} d_\mu^k(s_0) \sum_{a_0 \in \mathcal{A}'_{s_0}} \frac{\pi^k(a_0|s_0)}{C_k} \Delta = \frac{\sum_{s_0 \in \mathcal{S}_0} d_\mu^k(s_0) \sum_{a_0 \in \mathcal{A}'_{s_0}} \pi^k(a_0|s_0)\Delta}{\sum_{s_0 \in \mathcal{S}_0} \sum_{a_0 \in \mathcal{A}'_{s_0}} \pi^k(a_0|s_0)} \in [(1-\gamma)\tilde{\mu}\Delta, \Delta],$$

one has,

$$\frac{V^*(\mu) - V^{k+1}(\mu)}{V^*(\mu) - V^k(\mu)} \le \left(1 - \frac{\varepsilon}{\Delta\tilde{\mu}}\right)^{-1} \frac{\phi(\eta(-\Delta + \varepsilon))}{\phi(-\eta\varepsilon)} \left(\frac{\sigma|\mathcal{A}|}{(1-\gamma)^2\tilde{\mu}\Delta} + \frac{\delta\Delta \sum_{s_0 \in \mathcal{S}_0} \sum_{a_0 \in \mathcal{A}'_{s_0}} \pi^k(a_0|s_0)}{\Delta \sum_{s_0 \in \mathcal{S}_0} d_\mu^k(s_0) \sum_{a_0 \in \mathcal{A}'_{s_0}} \pi^k(a_0|s_0)} + 1\right)$$

$$\le \left(1 - \frac{\varepsilon}{\Delta\tilde{\mu}}\right)^{-1} \frac{\phi(\eta(-\Delta + \varepsilon))}{\phi(-\eta\varepsilon)} \left(\frac{\sigma|\mathcal{A}|}{(1-\gamma)^2\tilde{\mu}\Delta} + \frac{\delta}{(1-\gamma)\tilde{\mu}} + 1\right)$$

$$:= \zeta_1(\varepsilon, \delta, \sigma).$$

On the other hand, we have

$$\frac{V^*(\mu) - V^{k+1}(\mu)}{V^*(\mu) - V^k(\mu)} = \frac{\sum_s d_\mu^{k+1}(s) \sum_{a' \notin \mathcal{A}_s^*} \frac{\pi^{k+1}(a'|s)}{C_k} |A^*(s, a')|}{\sum_s d_\mu^k(s) \sum_{a' \notin \mathcal{A}_s^*} \frac{\pi^k(a'|s)}{C_k} |A^*(s, a')|}.$$

The denominator can be bounded as,

$$\sum_s d_\mu^k(s) \sum_{a' \notin \mathcal{A}_s^*} \frac{\pi^k(a'|s)}{C_k} |A^*(s, a')| = \sum_{s \notin \mathcal{S}_0} d_\mu^k(s) \sum_{a' \notin \mathcal{A}_s^*} \frac{\pi^k(a'|s)}{C_k} |A^*_{s, a'}|$$

$$+ \sum_{s_0} d_\mu^k(s_0) \sum_{a' \notin \left(\mathcal{A}_{s_0}^* \cup \mathcal{A}'_{s_0}\right)} \frac{\pi^k(a'|s_0)}{C_k} |A^*_{s_0, a'}|$$

$$+ \sum_{s_0} d_\mu^k(s_0) \sum_{a_0 \in \mathcal{A}'_{s_0}} \frac{\pi^k(a_0|s_0)}{C_k} \Delta$$

$$\le \frac{\sigma|\mathcal{A}|}{1-\gamma} + \sum_{s_0} d_\mu^k(s_0) \sum_{a_0 \in \mathcal{A}'_{s_0}} \frac{\pi^k(a_0|s_0)}{C_k} \Delta.$$

For the numerator, there holds

$$\sum_s d_\mu^{k+1}(s) \sum_{a' \notin \mathcal{A}_s^*} \frac{\pi^{k+1}(a'|s)}{C_k} |A^*(s,a')| \geq \sum_{s_0 \in \mathcal{S}_0} d_\mu^{k+1}(s_0) \sum_{a_0 \in \mathcal{A}'_{s_0}} \frac{\pi^{k+1}(a_0|s_0)}{C_k} \Delta$$

$$\geq \frac{\phi(-\eta(\Delta + \varepsilon))}{\phi(\eta\varepsilon)} \cdot \sum_{s_0 \in \mathcal{S}_0} (d_\mu^k(s_0) - \delta) \sum_{a_0 \in \mathcal{A}'_{s_0}} \frac{\pi^k(a_0|s_0)}{C_k} \Delta.$$

Consequently,

$$\frac{V^*(\mu) - V^{k+1}(\mu)}{V^*(\mu) - V^k(\mu)}$$

$$\geq \left( \frac{\phi(-\eta(\Delta + \varepsilon))}{\phi(\eta\varepsilon)} \right) \cdot \frac{\sum_{s_0 \in \mathcal{S}_0} (d_\mu^k(s_0) - \delta) \sum_{a_0 \in \mathcal{A}'_{s_0}} \frac{\pi^k(a_0|s_0)}{C_k} \Delta}{\frac{\sigma|\mathcal{A}|}{1-\gamma} + \sum_{s_0 \in \mathcal{S}_0} d_\mu^k(s_0) \sum_{a_0 \in \mathcal{A}'_{s_0}} \frac{\pi^k(a_0|s_0)}{C_k} \Delta}$$

$$= \left( \frac{\phi(-\eta(\Delta + \varepsilon))}{\phi(\eta\varepsilon)} \right) \cdot \left( \frac{\sum_{s_0 \in \mathcal{S}_0} d_\mu^k(s_0) \sum_{a_0 \in \mathcal{A}'_{s_0}} \frac{\pi^k(a_0|s_0)}{C_k} \Delta}{\frac{\sigma|\mathcal{A}|}{1-\gamma} + \sum_{s_0 \in \mathcal{S}_0} d_\mu^k(s_0) \sum_{a_0 \in \mathcal{A}'_{s_0}} \frac{\pi^k(a_0|s_0)}{C_k} \Delta} - \frac{\delta \sum_{s_0 \in \mathcal{S}_0} \sum_{a_0 \in \mathcal{A}'_{s_0}} \frac{\pi^k(a_0|s_0)}{C_k} \Delta}{\frac{\sigma|\mathcal{A}|}{1-\gamma} + \sum_{s_0 \in \mathcal{S}_0} d_\mu^k(s_0) \sum_{a_0 \in \mathcal{A}'_{s_0}} \frac{\pi^k(a_0|s_0)}{C_k} \Delta} \right)$$

$$\geq \left( \frac{\phi(-\eta(\Delta + \varepsilon))}{\phi(\eta\varepsilon)} \right) \cdot \left( \frac{1}{\frac{\sigma|\mathcal{A}|}{(1-\gamma)^2 \tilde{\mu}\Delta} + 1} - \frac{\delta\Delta}{\frac{\sigma|\mathcal{A}|}{1-\gamma} + (1-\gamma)\tilde{\mu}\Delta} \right)$$

$$:= \zeta_2(\varepsilon, \delta, \sigma).$$

As $\varepsilon \to 0, \delta \to 0, \sigma \to 0$, there holds $\zeta_1(\varepsilon, \delta, \sigma) \to \phi(-\eta\Delta)/\phi(0)$ and $\zeta_2(\varepsilon, \delta, \sigma) \to \phi(-\eta\Delta)/\phi(0)$. We finally have

$$\lim_{k \to \infty} \frac{V^*(\mu) - V^{k+1}(\mu)}{V^*(\mu) - V^k(\mu)} = \frac{\phi(-\eta\Delta)}{\phi(0)},$$

which completes the proof. □

# E PROOFS OF RESULTS IN SECTION 3.3

## E.1 PROOF OF LEMMA 3.1

The proof of Lemma 3.1 can be found in Lemma 2.12 of Liu et al. (2024b), so we omit the details here.

## E.2 PROOF OF LEMMA 3.2

According to the definition of $\phi$-update, it is obvious that

$$\sum_a \pi^{k+1}(a|s)A^k(s,a) = \frac{\mathbb{E}_{a \sim \pi^k(\cdot|s)}\left[A^k(s,a)\phi(\eta A^k(s,a))\right]}{\mathbb{E}_{a \sim \pi^k(\cdot|s)}\left[\phi(\eta A^k(s,a))\right]}.$$

From $\mathbb{E}_{a\sim\pi^k(\cdot|s)}\left[A^k(s,a)\right] = 0$ and the definition of $\xi^k(\cdot|s)$ in Lemma 3.2, we get

$$\pi^k(\mathcal{A}_s^k)\max_a A^k(s,a) + \left(1 - \pi_s^k\left(\mathcal{A}_s^k\right)\right)\mathbb{E}_{a'\sim\xi^k(\cdot|s)}\left[A^k(s,a')\right] = 0. \tag{14}$$

For the numerator (we use $A_{s,a}^k = A^k(s,a)$ for short),

$$\mathbb{E}_{a\sim\pi^k(\cdot|s)}\left[A_{s,a}^k\phi(\eta A_{s,a}^k)\right]$$
$$= \pi_s^k(\mathcal{A}_s^k)\phi(\eta\max_a A_{s,a}^k)\max_a A_{s,a}^k + \left(1-\pi_s^k(\mathcal{A}_s^k)\right)\mathbb{E}_{a'\sim\xi^k(\cdot|s)}\left[\phi(\eta A_{s,a'}^k)A_{s,a'}^k\right]$$
$$\geq \pi_s^k(\mathcal{A}_s^k)\phi(\eta\max_a A_{s,a}^k)\max_a A_{s,a}^k + \left(1-\pi_s^k(\mathcal{A}_s^k)\right)\mathbb{E}_{a'\sim\xi^k(\cdot|s)}\left[\phi(\eta A_{s,a'}^k)\right]\mathbb{E}_{a'\sim\xi^k(\cdot|s)}\left[A_{s,a'}^k\right]$$
$$= \pi^k(\mathcal{A}_s^k)\max_a A_{s,a}^k\left[\phi\left(\eta\max_a A_{s,a}^k\right) - \mathbb{E}_{a'\sim\xi^k(\cdot|s)}\left[\phi\left(\eta A_{s,a'}^k\right)\right]\right],$$

where the second inequality is from Lemma A.7 and the last line comes from the equation (14). For the denominator,

$$\mathbb{E}_{a\sim\pi^k(\cdot|s)}\left[\phi\left(\eta A_{s,a}^k\right)\right]$$
$$= \pi_s^k\left(\mathcal{A}_s^k\right)\phi\left(\eta\max_a A_{s,a}^k\right) + \left(1-\pi_s^k\left(\mathcal{A}_s^k\right)\right)\mathbb{E}_{a'\sim\xi^k(\cdot|s)}\left[\phi\left(\eta A_{s,a'}^k\right)\right]$$
$$= \pi_s^k\left(\mathcal{A}_s^k\right)\left[\phi\left(\eta\max_a A_{s,a}^k\right) - \mathbb{E}_{a'\sim\xi^k(\cdot|s)}\left[\phi\left(\eta A_{s,a'}^k\right)\right]\right] + \mathbb{E}_{a'\sim\xi^k(\cdot|s)}\left[\phi\left(\eta A_{s,a'}^k\right)\right].$$

Hence

$$\sum_a \pi^{k+1}(a|s)A^k(s,a) \geq \left[1 - \frac{1}{1 + \pi_s^k(\mathcal{A}_s^k)\left(\dfrac{\phi\left(\eta\max_a A^k(s,a)\right)}{\mathbb{E}_{a'\sim\xi^k(\cdot|s)}\left(\phi\left(\eta A_{s,a'}^k\right)\right)} - 1\right)}\right]\max_a A^k(s,a)$$

$$= \left[1 - \frac{1}{1 + \pi_s^k(\mathcal{A}_s^k)\left(\Delta_{\phi,s}^k(\eta) - 1\right)}\right]\max_a A^k(s,a).$$

### E.3 PROOF OF THEOREM 3.5

Combining Lemma 3.1 and Lemma 3.2 together yields this result.

### E.4 A BANDIT EXAMPLE FOR THE TIGHTNESS OF THEOREM 3.5

Consider the following bandit problem:

$$\mathcal{S} = \{s\}, \quad \mathcal{A} = \{a_1, a_2\}, \quad r(s, a_1) = 1, \quad r(s, a_2) = 0.$$

Under this problem setting, it is clear that for any $\pi$ ($\pi_{a_i}$ is short for $\pi(a_i|s)$)

$$Q^\pi(s, a_1) = 1, \quad Q^\pi(s, a_2) = 0, \quad V^\pi(s) = \pi_{a_1},$$
$$A^\pi(s, a_1) = 1 - \pi_{a_1}, \quad A^\pi(s, a_2) = -\pi_{a_1}.$$

It is also clear that $a_1$ is the optimal action and $V^*(s) = 1$.

The $\phi$-update on action $a_1$ is given by (the subscript $s$ in $\eta_s^k$ is also omitted)

$$\pi_{a_1}^{k+1} = \frac{\pi_{a_1}^k \phi\left(\eta^k\left(1 - \pi_{a_1}^k\right)\right)}{Z^k},$$

where

$$Z^k = \pi_{a_1}^k \phi\left(\eta^k\left(1 - \pi_{a_1}^k\right)\right) + (1 - \pi_{a_1}^k)\phi\left(-\eta^k \pi_{a_1}^k\right).$$

It can be verified directly that

$$\left(V^*(s) - V^{k+1}(s)\right)\left(1 + \pi_{a_1}^k\left(\frac{\phi(\eta^k(1 - \pi_{a_1}^k))}{\phi(-\eta^k\pi_{a_1}^k)} - 1\right)\right)$$

$$= \left(1 - \pi_{a_1}^{k+1}\right)\left(1 + \pi_{a_1}^k\left(\frac{\phi(\eta^k(1 - \pi_{a_1}^k))}{\phi(-\eta^k\pi_{a_1}^k)} - 1\right)\right)$$

$$= 1 - \pi_{a_1}^k$$

$$= V^*(s) - V^k(s).$$

Consequently,

$$V^*(s) - V^k(s) = \left(V^*(s) - V^0(s)\right)\prod_{t=0}^{k-1}\left[1 + \pi_{a_1}^t\left(\frac{\phi(\eta^t(1 - \pi_{a_1}^t))}{\phi(-\eta^t\pi_{a_1}^t)} - 1\right)\right]^{-1}.$$

Noting that $\gamma = 0$ and $\pi_s^t(\mathcal{A}_s^t) = \pi_{a_1}^t$, it shows that the bound in Theorem 3.5 holds with equality for this example.

### E.5 Proof of Theorem 3.6

For sufficiently small $\varepsilon > 0$, by Theorem 3.5 we have

$$\forall\, k \leq T(\varepsilon): \quad \left\|V^* - V^k\right\|_\infty \leq \left\|V^* - V^0\right\|_\infty \prod_{t=0}^{k-1}\left(1 - (1 - \gamma)\left[1 - \frac{1}{1 + D_t}\right]\right)$$

$$\leq \left\|V^* - V^0\right\|_\infty \kappa(\varepsilon)^k.$$

By Theorem 3.2 (see Eq (11)) we have

$$\forall\, k \geq T(\varepsilon): \quad \left\|V^* - V^k\right\|_\infty \leq \frac{1}{(1 - \gamma)\tilde{\mu}^2}\left\|V^* - V^{T(\varepsilon)}\right\|_\infty \cdot \rho_1\left(\eta, \Delta, \varepsilon\right)^{k - T(\varepsilon)}.$$

Combining them we get

$$\forall\, k \in \mathbb{N}^+: \quad \left\|V^* - V^k\right\|_\infty \leq \frac{\left\|V^* - V^0\right\|_\infty}{(1 - \gamma)\tilde{\mu}^2} \cdot \left[\max\left\{\kappa(\varepsilon),\ \rho_1\left(\eta, \Delta, \varepsilon\right)\right\}\right]^k$$

$$= \frac{\left\|V^* - V^0\right\|_\infty}{(1 - \gamma)\tilde{\mu}^2} \cdot \rho(\varepsilon)^k.$$

## F  More examples of $\phi$-update

### F.1  Variants of softmax NPG

Recall that softmax NPG is a $\phi$-update instance with $\phi = \exp(\cdot)$. It is natural to generalize it to an exponential family of $\phi$. For instance, it is direct to consider the following exponential family $\phi$:

$$\phi(t) = \exp(\sqrt[q]{(t)^p})$$

for some positive and odd $p$, $q$[1]. When $p \geq q$, it is clear $\phi(t)$ satisfies the assumptions (I), (II), (III). When $p < q$, $\phi(t)$ is not differentiable around $0$, so we consider the following piece-wise variant:

$$\phi(t) = \begin{cases} \exp(\sqrt[q]{(t)^p}) & \text{if } t \in (-\infty, -\delta] \cup [\delta, +\infty), \\ \exp(\delta^{p/q-1} \cdot t) & \text{if } t \in (-\delta, \delta) \end{cases} \tag{15}$$

with some small constant $\delta > 0$. It can be easily verified that $\dfrac{d}{dt} \log \phi(t) = \delta^{p/q} - 1$ when $t \in (-\delta, \delta)$, thus the assumptions (I), (II), (III) hold.

The convergence results of $\phi$-update for difference $p$ and $q$ (when $p < q$, we use (15)) are presented in Figure 4(a). The theoretical results of Theorem 3.4 and 3.6 imply that the convergence rate becomes smaller when $\phi(t)$ changes rapidly around $t = 0$. Noting that when $p < q$ and $\delta$ is small, $\phi(t)$ in the 0-neighbourhood is "stretched", and the stretch intensity becomes stronger as $p/q$ and $\delta$ are smaller. The simulations show that under the exact gradient, $\phi$-update enjoys a better tail convergence than softmax NPG when $p < q$.

### F.2 POLYNOMIAL UPDATE

Recall the policy update form of Hadamard PG, which is a special case of $\phi$-update with $\phi(t) = (1 + 2t)^2$. Thus it is natural to generalize it to a family of $\phi$-updates (coined as polynomial update here and denoted by $\text{Poly}(p)$ for different $p \in \mathbb{Z}^+$) with $\phi$ being given by

$$\phi(t) = (1 + pt)^p.$$

It is clear that $\phi(t)$ satisfies (I), (II), (III) with $L = 1/p$. It can also be observed that $\phi(t) = (1 + pt)^p$ stretches around $t = 0$ as $p$ increases.

**Remark F.1.** *Recall the escort parameterization in* Mei et al. (2020a) *with hyperparameter p,*

$$\pi_\theta(a|s) = \frac{|\theta_{s,a}|^p}{\sum_a |\theta_{s,a}|^p} = \frac{|\theta_{s,a}|^p}{\|\theta_s\|_p^p}.$$

*Escort PG updates the parameters by*

$$\forall s \in \mathcal{S}, \ a \in \mathcal{A}: \quad \theta_{s,a}^{k+1} = \theta_{s,a}^k + \eta \cdot \|\theta_s^k\|_p^2 \cdot \frac{\partial V^k(\mu)}{\partial \theta_{s,a}^k}. \tag{16}$$

*Under escort parameterization, it can be verified that* $\text{Poly}(p)$ *can be implemented by modifying* (16) *to*

$$\forall s \in \mathcal{S}, \ a \in \mathcal{A}: \quad \theta_{s,a}^{k+1} = \theta_{s,a}^k + \eta \cdot \|\theta_s^k\|_p^p \cdot (\theta_{s,a}^k)^{-(p-2)} \cdot (\text{sgn}(\theta_{s,a}^k))^p \cdot \frac{\partial V^k(\mu)}{\partial \theta_{s,a}^k},$$

*where* $\text{sgn}(\cdot)$ *is the sign function. In Figure 4(b) we compare* $\text{Poly}(p)$ *and escort PG with different selections of p.*

## G PRELIMINARY NUMERICAL RESULTS ON $\phi$-UPDATE UNDER NEURAL NETWORK PARAMETERIZATION

We have conducted some preliminary numerical experiments on $\phi$-update under the neural network parameterization for the exponential family presented in Section F.1. Three environments from MuJoCo are tested,

---

[1] We use $\sqrt[q]{(\cdot)^p}$ instead of $(\cdot)^{p/q}$ to avoid the ambiguity of the power function. For instance $p = 3$, $q = 5$ and $p = 6$, $q = 10$ are different under $\sqrt[q]{(\cdot)^p}$ but equivalent under $(\cdot)^{p/q}$.

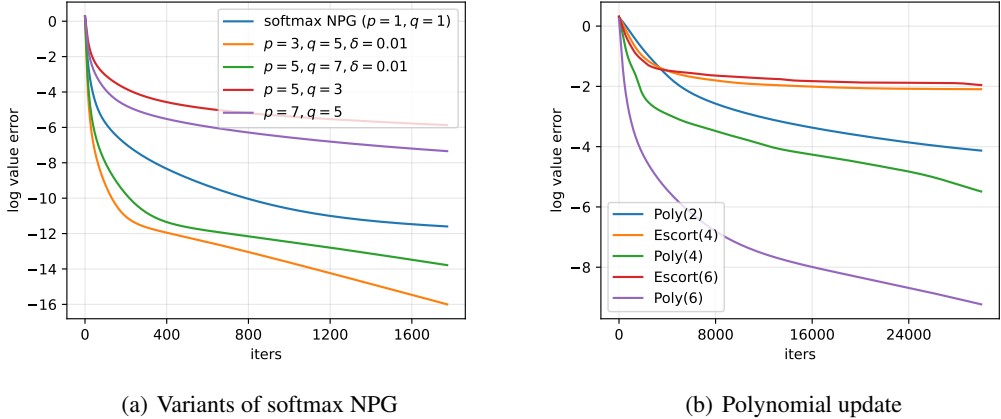

(a) Variants of softmax NPG

(b) Polynomial update

Figure 4: Simulation results on random MDP problem (with the same setting to the experiments presented in Figure 3). The step-size is set to be $\eta = 1$ for (a). For (b), the step-size is state-dependent following Mei et al. (2020a) and set to be $0.01 \times \left\| \theta_s^k \right\|_p^2$ for escort PG, and is set to be $0.01$ for $\mathrm{Poly}(p)$. Note that $\mathrm{Poly}(2)$ is equivalent to Hadamard PG or escort PG with $p = 2$.

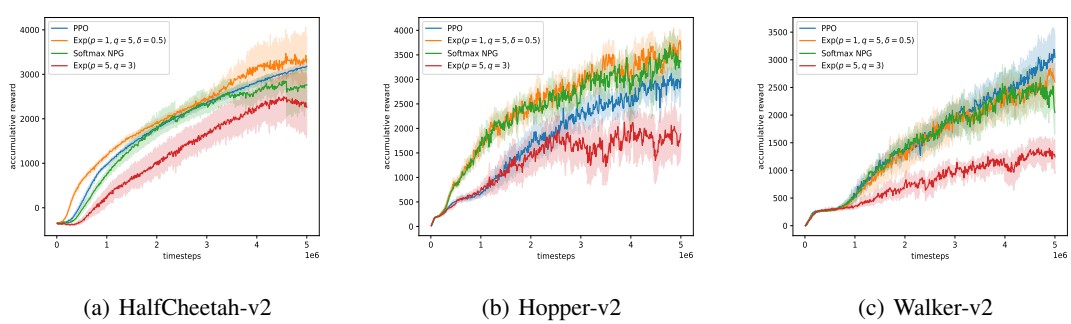

(a) HalfCheetah-v2

(b) Hopper-v2

(c) Walker-v2

Figure 5: Preliminary numerical results on $\phi$-update under neural network parameterization.

and results are presented Figure 5. For each experiment, we compute the mean and the standard deviation of the final accumulative reward across 10 different random seeds. The policy neural network is a two-layers MLP with 64 units per layer, and the timesteps are 5 millions for each experiment.

Note that the numerical results here are only intended to illustrate implementation feasibility of $\phi$-update, rather than to claim its state-of-the-art results. In fact, comprehensive empirical evaluations of $\phi$-update is beyond the scope of this paper, especially on how to choose $\phi$ for different problems, which will be left for future work.

# H    CONVERGENCE RESULTS FOR NON-CONSTANT STEP SIZE

In the following, we present the convergence results for the adaptive step size sequence $\{\eta_s^k\}$. The proofs are overall similar to the case of constant step size (Appendix C, D, E). Thus, we do not provide complete proof details, but only point out the proof differences.

**Proposition H.1.** *Suppose the assumptions* (I), (II) *hold. For any positive step-size sequence* $\{\eta_s^k\} > 0$, *$\phi$-update has non-negative improvement at every state, i.e.,* $\sum_a \pi^{k+1}(a|s)A^k(s,a) \geq 0$ *for any $s$ and $k$. Furthermore, we have* $V^{k+1}(s) \geq V^k(s)$ *for every* $s \in \mathcal{S}$.

*Proof.* The proof is the same with that for Proposition 3.1 as $\eta_s^k > 0$. $\qquad\square$

**Theorem H.1** (Global Convergence). *Suppose the assumptions* (I), (II) *hold. For a step-size sequence* $\{\eta_s^k\}$ *such that* $\inf_{s,k} \eta_s^k > 0$, *the value function generated by $\phi$-update method converges to the optimal value, i.e.,* $V^\infty(s) = V^*(s)$ *for all state $s$.*

*Proof.* First note that the proof of Lemma C.1 remains unchanged for non-constant step-sizes $\{\eta_s^k\}$.

Let $c_3 := \inf_{s_0,k} \eta_{s_0}^k > 0$ where $s_0$ is the state such that $I_{s_0}^+$ is non-empty. By the same computation,

$$
\sum_{a \in \mathcal{A}} \pi^{k+1}(a|s_0)A^k(s_0,a)
$$

$$
\geq \frac{1}{2Z_{s_0}^k} \sum_{a \in I_{s_0}^+} \sum_{a' \in I_{s_0}^-} \pi^k(a|s_0)\pi^k(a'|s_0) \left[ \left(A^k(s_0,a) - A^k(s_0,a')\right) \left(\phi\left(\eta_{s_0}^k A^k(s_0,a)\right) - \phi\left(\eta_{s_0}^k A^k(s_0,a')\right)\right) \right]
$$

$$
\geq \frac{1}{2Z_{s_0}^k} \sum_{a \in I_{s_0}^+} \sum_{a' \in I_{s_0}^-} \pi^k(a|s_0)\pi^k(a'|s_0) \left[ (\varepsilon - (-\varepsilon)) \left(\phi\left(\eta_{s_0}^k A^k(s_0,a)\right) - \phi(-c_3\varepsilon)\right) \right]
$$

$$
\geq \frac{c_2\varepsilon}{Z_{s_0}^k} \sum_{a \in I_{s_0}^+} \pi^k(a|s_0) \left(\phi\left(\eta_{s_0}^k A^k(s_0,a)\right) - \phi(-c_3\varepsilon)\right)
$$

$$
= \frac{c_2\varepsilon}{Z_{s_0}^k} \left(l_{s_0}^k\left(I_{s_0}^+\right) - \pi_{s_0}^k\left(I_{s_0}^+\right)\phi(-c_3\varepsilon)\right).
$$

Similarly, one has

$$
\sum_{a \in \mathcal{A}} \pi^{k+1}(a|s_0)A^k(s_0,a) \geq \varepsilon \cdot c_1 \cdot c_2 \cdot \left(1 - \frac{\phi(-c_3\varepsilon)}{\phi(c_3\varepsilon)}\right) > 0,
$$

which also raises an contradiction. Hence the proof of Theorem 3.1 can be proceeded for the non-constant step-sizes provided $\inf_{s,k} \eta_s^k > 0$. $\qquad\square$

**Lemma H.1.** *Suppose the assumptions* (I), (II) *hold. For small enough* $\varepsilon > 0$, *when $k \geq T(\varepsilon)$ we have*

$$
\forall\, s \in \tilde{\mathcal{S}},\ a' \notin \mathcal{A}_s^* : \quad \frac{\pi^{k+1}(a'|s)}{\pi^k(a'|s)} \leq \frac{\phi\left(-\eta_s^k(\Delta_s - \varepsilon)\right)}{\phi\left(-\eta_s^k\varepsilon\right)} \cdot \left(1 - \frac{\varepsilon}{\Delta}\right)^{-1} := \rho_1(\eta_s^k, \Delta_s, \varepsilon), \tag{17}
$$

*and*

$$
\forall\, s \in \tilde{\mathcal{S}},\ a' \in \mathcal{A}_s' : \quad \frac{\pi^{k+1}(a'|s)}{\pi^k(a'|s)} \geq \frac{\phi\left(-\eta_s^k(\Delta_s + \varepsilon)\right)}{\phi\left(\eta_s^k\varepsilon\right)} := \rho_2(\eta_s^k, \Delta_s, \varepsilon). \tag{18}
$$

*Proof.* The proof remains the same with that for Lemma B.1. □

**Lemma H.2.** *Suppose the assumptions* (I), (II) *hold. Denote* $\varepsilon_k = \left\|V^* - V^k\right\|_\infty$. *For any* $s \in \mathcal{S}$ *and the optimal action* $a^* \in \mathcal{A}_s^*$, *when* $\varepsilon_k < \Delta$ *we have*

$$\frac{\pi^{k+1}(a^*|s)}{\pi^k(a^*|s)} \geq \frac{\phi(-\eta_s^k \varepsilon_k)}{\phi(\eta_s^k \varepsilon_k) + \frac{\varepsilon_k}{\Delta}\phi(-\eta_s^k(\Delta - \varepsilon_k))},$$

*and*

$$\frac{\pi^{k+1}(a^*|s)}{\pi^k(a^*|s)} \leq \frac{\phi(\eta_s^k \varepsilon_k)}{\left(1 - \frac{\varepsilon_k}{\Delta}\right)\phi(-\eta_s^k \varepsilon_k)}.$$

*Proof.* The proof remains the same with that for Lemma B.2. □

**Theorem H.2** (Local Linear Upper Bound). *Suppose the assumptions* (I), (II) *hold. Assume that* $\inf_{s,k} \eta_s^k > 0$. *Then there exists a small enough* $\varepsilon > 0$ *such that* $\sup_{s,k} \rho_1(\eta_s^k, \Delta, \varepsilon) < 1$, *and a time* $T(\varepsilon)$ *such that the values generated by* $\phi$*-update method satisfy*

$$\forall k \geq T(\varepsilon) : \quad \left\|V^* - V^k\right\|_\infty \leq \frac{\left[\sup_{s,k} \rho_1\left(\eta_s^k, \Delta, \varepsilon\right)\right]^{(k-T(\varepsilon))}}{(1-\gamma)^2 \tilde{\mu}^2}.$$

*Proof.* By Lemma H.1, we have

$$\forall k \geq T(\varepsilon), \ s \in \tilde{\mathcal{S}}, \ a \notin \mathcal{A}_s^* : \quad \pi^k(a|s) \leq \rho_1(\eta_s^{k-1}, \Delta, \varepsilon)\pi^{k-1}(a|s)$$
$$\leq \dots$$
$$\leq \left[\prod_{t=T(\varepsilon)}^{k-1} \rho_1(\eta_s^t, \Delta, \varepsilon)\right] \cdot \pi^{T(\varepsilon)}(a|s)$$
$$\leq \left[\sup_{s,k} \rho_1\left(\eta_s^k, \Delta, \varepsilon\right)\right]^{(k-T(\varepsilon))} \cdot \pi^{T(\varepsilon)}(a|s).$$

Then the proof in Theorem 3.2 can be proceeded similarly as the constant step size case. □

**Theorem H.3** (Policy Convergence). *Suppose the assumptions* (I), (II), (III) *hold. With the step-size sequence* $\{\eta_s^k\}$ *satisfying* $\inf_{s,k} \eta_s^k > 0$ *and* $\sup_{s,k} \eta_s^k < +\infty$, *the policy generated by* $\phi$*-update converges to some optimal policy* $\pi^*$, *i.e. the sequence* $\{\pi^k(a|s)\}_k$ *converges for any* $s \in \mathcal{S}$ *and* $a \in \mathcal{A}$.

*Proof.* Following the same procedure of the proof for Theorem 3.3, by selecting small enough $\varepsilon$ and $\varepsilon_0$ such that $\sup_{s,k} \rho_1(\eta_s^k, \Delta, \varepsilon) \leq \sup_{s,k} \rho_1(\eta_s^k, \Delta, \varepsilon_0) < 1$, one can get

$$\sum_{k=T(\varepsilon)}^\infty \left|\log \pi^{k+1}(a^*|s) - \log \pi^k(a^*|s)\right| \leq \frac{\frac{1}{\Delta} + 2c\sup_{s,k} \eta_s^k}{(1-\gamma)\tilde{\mu}^2} \frac{2\varepsilon}{1 - \sup_{s,k} \rho_1(\eta_s^k, \Delta, \varepsilon_0)} := \tilde{C}_0 \varepsilon.$$

Hence $\{\log \pi^k(a^*|s)\}_k$ is still a Cauchy sequence, implying that $\pi^k(a^*|s)$ converges. □

**Theorem H.4** (Exact Asymptotic Linear Convergence). *Suppose the assumptions* (I), (II), (III) *hold. Assume* $\eta_s^\infty := \lim_k \eta_s^k$ *exists for all* $s \in \mathcal{S}$*. Then*

$$\lim_{k \to \infty} \frac{V^*(\mu) - V^{k+1}(\mu)}{V^*(\mu) - V^k(\mu)} = \frac{\phi(- \min_{s,a}\{\eta_s^\infty |A^*(s,a)|\})}{\phi(0)}.$$

*Proof.* One can follow the same procedure as in the proof of Theorem 3.4 to get this result by further considering the local region of $\eta_s^\infty$. That is, consider the following local region:

$$\forall s \in \mathcal{S}, \ k \geq T(\varepsilon, \delta, \sigma, \xi) : \quad \left| \frac{d_\mu^{k+1}(s)}{d_\mu^k(s)} - 1 \right| \leq \delta, \quad \left| \frac{d_\mu^k(s)}{d_\mu^{k+1}(s)} - 1 \right| \leq \delta,$$

$$\left\| A^* - A^k \right\|_\infty \leq \varepsilon, \quad \frac{\pi^k(a|s)}{C_k} \leq \sigma, \quad \left| \eta_s^k - \eta_s^\infty \right| \leq \xi$$

where $\varepsilon, \delta, \sigma, \xi$ are all sufficiently small constants. $\qquad\square$

**Lemma H.3** (Improvement Lower Bound). *Suppose the assumptions* (I), (II) *hold. With positive step-size sequence* $\{\eta_s^k\}$*, the* $\phi$*-update improvement of state* $s$ *satisfies*

$$\sum_a \pi^{k+1}(a|s)A^k(s,a) \geq \left[ 1 - \frac{1}{1 + \pi_s^k(\mathcal{A}_s^k)\left(\Delta_{\phi,s}^k(\eta_s^k) - 1\right)} \right] \cdot \max_a A^k(s,a),$$

*where*

$$\Delta_{\phi,s}^k(\eta_s^k) := \frac{\phi\left(\eta_s^k \max_a A^k(s,a)\right)}{\mathbb{E}_{a' \sim \xi^k(\cdot|s)}\left[\phi\left(\eta_s^k A^k(s,a')\right)\right]} \quad with \quad \xi^k(a|s) = \begin{cases} 0 & \text{if } a \in \mathcal{A}_s^k, \\ \pi^k(a|s)/(1 - \pi_s^k(\mathcal{A}_s^k)) & \text{if } a \notin \mathcal{A}_s^k. \end{cases}$$

*Proof.* The proof remains the same with that for Lemma 3.2. $\qquad\square$

**Theorem H.5.** *Suppose the assumptions* (I), (II) *hold. With positive step-size sequence* $\{\eta_s^k\}$*, the value function generated by* $\phi$*-update satisfies*

$$\forall k \in \mathbb{N}^+ : \quad \left\| V^* - V^k \right\|_\infty \leq \left\| V^* - V^0 \right\|_\infty \prod_{t=0}^{k-1} \left( 1 - (1 - \gamma)\left[ 1 - \frac{1}{1 + D_t} \right] \right),$$

*where* $D_t := \min_{s \in \tilde{\mathcal{S}}_t} \left\{ \pi_s^t(\mathcal{A}_s^t)\left(\Delta_{\phi,s}^t(\eta_s^t) - 1\right) \right\}$ *and* $\tilde{\mathcal{S}}_t := \{s \in \mathcal{S} : \mathcal{A}_s^t \neq \mathcal{A}\}$*.*

*Proof.* The proof remains the same with that for Theorem 3.5. $\qquad\square$

**Theorem H.6** (Global Linear Convergence). *Suppose the assumptions* (I), (II) *hold. Assume that* $\inf_{s,k} \eta_s^k > 0$*. Define*

$$\rho(\varepsilon) = \max \left\{ \kappa(\varepsilon), \ \sup_{s,k} \rho_1\left(\eta_s^k, \Delta, \varepsilon\right) \right\}$$

*where*

$$\kappa(\varepsilon) = \max_{k \leq T(\varepsilon)} \left\{ 1 - (1 - \gamma)\left[ 1 - \frac{1}{1 + D_t} \right] \right\}.$$

*There exists* $\varepsilon > 0$ *such that* $\rho(\varepsilon) < 1$ *and*

$$\forall k \in \mathbb{N}^+ : \quad \left\| V^* - V^k \right\|_\infty \leq \frac{\left\| V^* - V^0 \right\|_\infty}{(1 - \gamma)\tilde{\mu}^2} \cdot \rho(\varepsilon)^k.$$

*Proof.* Similar to the proof of Theorem 3.6, but instead use Theorem H.2 and Theorem H.5. $\qquad\square$

