# OpenReview forum: "$\phi$-Update: A Class of Policy Update Methods with Policy Convergence Guarantee"
_ICLR.cc/2025/Conference — ICLR 2025 Poster_

### Official Review · Reviewer_NGsf · 2024-10-22

**Soundness:** 3
**Presentation:** 3
**Contribution:** 3
**Rating:** 6
**Confidence:** 3

**Summary:**

This paper introduces a general policy update, called the $\phi$-update, inspired by softmax NPG and Hadamard PG. Under the assumption of exact gradients, convergence is analyzed in the tabular setting. First, global asymptotic convergence is shown for the value function, as well as for the policy itself. Then, a linear rate of convergence is established. The authors provide a tabular numerical toy example to verify the tight theoretical convergence rate under exact gradients and describe how to extend the theory of $\phi$-updates to general non-tabular parameterizations.

**Strengths:**

**S1:**	Well-motivated new policy update.

**S2:**	Solid theoretical analysis in the exact gradient setting, both asymptotic and non-asymptotic, although I have a question regarding the linear convergence rate (see below).

**S3:**	The exact gradient rates are tight, which is verified in numerical examples.

**Weaknesses:**

**W1:** The constant $\rho(\epsilon)$ is not exactly stated in terms of model parameters. How close is this constant to $1$?

**W2:** Analysis limited to exact gradient case, stochastic analysis would be interesting.

**W3:** The authors present examples in the tabular case. It would be nice to also analyse the generalised variant presented in Sec. 4.2. It remains unclear if this version works in practical applications.

Although I understand that this is a theoretical paper which leaves stochastic analysis and experiments to future work, it would be nice to have at least one (toy) stochastic application that verifies practical performance

**Minors:**

Typo in the definition of $\kappa(\epsilon)$? It should be $\max_{t\leq T(\epsilon)}$ as there is no $k$ apprearing within the max.

**Questions:**

**Q1:** Can you give a more precise description of the constant $\rho(\epsilon)$, in terms of model parameters like $\gamma$, $|\mathcal S|$ or $|\mathcal A|$? You stated yourself that in (Mei et al., 2020b) a constant $c$ appears in the softmax convergence rate which can be extremely small. (Mei et al., 2020b) state a sublinear convergence rate which can have very poor performance with respect to the discount factor (see [1]) and it would be nice to know if such a counter example can also appear for your algorithm. The term in the definition of $D_t$ looks very similar to the "bad" constant $c$ in (Mei et al., 2020b).

**Q2:** The error curves in the simulations in Fig. 2 and 3. look very smooth. Did you use the transition probabilities to calculate the exact $\phi$-updates or did you use simulations?



[1] *Li, Gen, Yuting Wei, Yuejie Chi, and Yuxin Chen. “Softmax Policy Gradient Methods Can Take Exponential Time to Converge.” Mathematical Programming 201, no. 1–2 (September 2023): 707–802. https://doi.org/10.1007/s10107-022-01920-6.*

---

> ### Author Response · Authors · 2024-11-17
> **Response to Reviewer NGsf**
>
> Thank you for pointing out the typo. We will fix it in the revision.
>
> **Response to W1 & Q1:** The convergence rate $\rho(\varepsilon)$ is the maximum of two rates, $\rho_1$ and $\kappa(\varepsilon)$ from two phases of the iterations. The first rate $\rho_1(\eta, \Delta,\varepsilon)$ comes from the local convergence phase (Theorem 3.2), which can be explicitly expressed in terms of $\phi$, $\eta$, $\Delta$ and $\varepsilon$. For the second rate $\kappa(\varepsilon)$, it comes from the global dynamic convergence (Theorem 3.5) for the initial finite number of iterations and depends on $D_t$. As we have provided a bandit example in Appendix E.4 to show that the error bound in Theorem 3.5 is tight, the global convergence necessarily depends on the term $D_t$, which cannot be explicitly described by model parameters.\
> Consider again the bandit problem in Appendix E.4, where the value error of this problem is explicitly shown in Line 1334. If one set the initial policy $\pi^0_{a_1}$ to be very close to $0$, the convergence rate in the early iterations will be extremely small, so the global convergence rate $\rho(\varepsilon)$ will be close to $1$. Noting that the value error in Line 1334 is exact, it is impossible to improve $\rho(\varepsilon)$ for such policy initialization.\
> Despite that $\rho(\varepsilon)$ cannot be explicitly expressed in term of the model parameters, the global convergence result is still interesting in our opinion. In optimization, global linear convergence usually can only be established for strongly-convex (or strongly-concave) functions, with the convergence rate relying on the parameter controlling the strongly convex property. For problems without this property, it is typically difficult to establish the global linear convergence.  However, we show that for the RL problem (even non-convex and non-concave), the establishment of linear convergence is possible  by leveraging the particular MDP structure even though it is difficult to give a concise expression for the convergence rate due to the lack of certain uniform property. It is worth noting that most of the existing global linear convergence results are either based on entropy regularization or using exponentially increasing step sizes.
>
> **Response to W2 & W3:** We agree that the analysis under stochastic setting is interesting which is left in the future work. In this paper, we focus on the theory for the exact gradient setting, especially having established the policy convergence (without regularization) and the exact asymptotic convergence rate which are totally new, to the best of our knowledge.\
> The $\phi$-update can be implemented with general parameterizations under the stochastic setting. We have tentatively conducted some preliminary numerical experiments under parameterizations with neural networks, and results are presented in the table in our general response to all reviewers above. The experiment results show that $\phi$-update indeed works with some specific $\phi$ under stochastic applications. Because there are still a lot work to do to comprehensively evaluate the empirical performance of $\phi$-update (e.g., how to choose $\phi$ for difference problems), we prefer to focus on the theory in this paper, and not to include related numerical results.
>
> **Response to Q2:** Since the purpose of these experiments is to empirically verify the results in Theorem 3.3 and Theorem 3.4, the simulations in Figure 2 and 3 are conducted under the exact gradient setting, and we use the transition probabilities to calculate the exact advantage functions.

---

> ### Comment · Reviewer_NGsf · 2024-11-21
>
> Thank you for addressing my concerns and for providing such a detailed response. I sincerely appreciate the effort you put into clarifying key aspects of the work.
>
> I fully agree that advancing the theoretical understanding of the convergence properties of policy gradient methods is both important and valuable to the community. While I still support the novelty and contributions of your paper, I would like to explain why I maintain my original score rather than raising it.
>
> - *Numerical Examples:* I appreciate the inclusion of numerical examples demonstrating the practical feasibility of your method. I strongly encourage you to include these examples in the appendix of the final version, along with an explanation (similar to what you provided in the response) that these examples are intended to illustrate implementation feasibility rather than serve as comprehensive performance benchmarks. Additionally, open-sourcing the code would further enhance the accessibility and utility of your work for the broader research community.
> - *Discussion of Imprecise Numerical Rates:* While the numerical convergence rates presented are tight, they lack precision, and I believe it is essential to discuss the implications of this imprecision in the paper. For example, the impreciseness of the constant "c" derived by Mei et al. (2020) in vanilla policy gradient methods is not explicitly addressed, and I suspect this may lead to a lack of awareness in the community about potential issues arising from this (as noted in Li et al. (2023) in my review). Though such issues might be rare in practice, they are still meaningful and merit discussion. I also suggest referencing recent work, such as Klein et al. (2024), which proposes new algorithms to address similar challenges, to contextualize your contribution in light of ongoing developments.
>
> In conclusion, I continue to value the originality of the $\phi$-update approach and its potential impact, and I support accepting the paper to ICLR. I would raise my score to 7, but since 7 is not an option and due to the constant issue, which is often swept under the rug and not precisely addressed, I cannot give it an 8.
>
> S. Klein, X. Zhang, T.Başar, S. Weissmann, L. Döring, *„Structure Matters: Dynamic Policy Gradient“*, 2024.

---

> > ### Author Response · Authors · 2024-11-22
> >
> > Thank you for the supportive comments. Though the bandit example basically implies that the rate in the global linear convergence is roughly tight, we agree that it would be  desirable to provide a more quantitative characterization. Moreover, it is interesting to see whether there is a worst case example which shows the rate can be exponentially small, as in [1].  Thank you for providing us the new and related reference, and we will be happy to cite it.  It is also interesting to see whether we can use the techniques therein to  accelerate $\phi$-update. These discussions have been summarized in Remark 3.8 in the paper.
> >
> >
> > About the experiments, we have followed the suggestion and include the preliminary results (in terms of plots instead of table for more clarity) in Appendix G. We can open-source the codes in the future after they are further organized and optimized.
> >
> > **References:**
> >
> > [1] Li, Gen, et al. *Softmax policy gradient methods can take exponential time to convergence.* Mathematical Programming, 2023.

---

### Official Review · Reviewer_7X97 · 2024-11-04

**Soundness:** 4
**Presentation:** 3
**Contribution:** 4
**Rating:** 8
**Confidence:** 4

**Summary:**

phi-update

Summary: The paper proposes a new "general" class of update rules for "policy optimization" to find optimal policies in Markov Decision Processes. The new update rules takes the form $\pi'(a|s) \propto \pi(a|s) \phi( \eta(s)A^\pi(s,a))$, where $\pi'$ is the new policy, $\pi$ is the previous policy and $A^\pi$ is the advantage function of $\pi$, while $\eta(s)$ is a step-size that may depend on the state (but not on the actions), and $\phi$ is a "transfer" function that maps reals to reals. By choosing $\phi(t)=\exp(t)$ we get back the update of the "natural policy gradient" method, while $\phi(t) = (1+2t)^2 we get back the update rule of what is known as "Hadamard PG" (which is related to the "escort transform" based updates of some other previous work). In a nutshell, the new update rule unifies some previous update rules. The main result in the paper is the convergence of policies to some optimal policy and the exact asymptotic rate of value subotimality, while the stepsizes are allowed to be very general (as long as they are bounded away from zero; including adaptive stepsizes).

Significance: There is considerable interest in the community to gain a deeper understanding of how to design effective and efficient algorithms and one line of active research looks at variations of "policy gradient" methods, or, as done in this paper, at variations of policy update rules. The contribution of this paper should thus be interpreted in this context. I think the question asked is reasonable and it deepens our understanding of policy update rule design. The results go beyond previous results in the area in a number of ways: The authors manage to remove the (annoying and, as it turns out, unnecessary) condition that the optimal actions are unique at all states. The asymptotic convergence rate result is also a nice contribution.

Soundness: I looked at the proofs at various places and did not find any problems with them and the results do look reasonable to me. I believe the results are sound.

Related work: The authors do a good job at discussing related work, except for two things: (1) the relation to the escort transform based work remains a bit unclear. (2) I feel that maybe the novelty of the proof techniques is overclaimed on the basis that the update is not given in the parameter space. However, this was only because the authors chose not to consider the parameter space updates. With a softmax parameterization, an equivalent form for the update is $\theta'(s,a)=\theta(s,a) + \log \phi( \eta(s) A^\pi(s,a))$. This also shows that the function $\psi(t) = \log(\phi(t))$ is perhaps even more fundamental to these update rules (condition (III) in the paper is indeed a condition on the rate of growth of this function in the vicinity of zero, which I feel demistifies this condition quite a bit).

Presentation: There are quite a few typos, grammatical problems (I will add a list to the end of my review in the coming days to help the authors correct the ones I found). There are also symbols used in the main text (e.g., $A^k$, $\mathcal{A}_s^k$ used in line 322, ..) with no definitions given for them as far as I could see (line 133 was maybe an attempt to define all these quantities; but it did not..). Nevertheless, these problems are relatively minor  and can be fixed in a revision easily. Also, I found that some results added little if anything. E.g., Theorems 3.5, and 3.6 are full of algorithm dependent parameters and thus have no clear interpretation (other than they also state an asymptotic rate). Thus, should they be part of the main text?

**Strengths:**

Interesting problem, interesting results, new insightful proofs (avoiding uniqueness of optimal action!, generalization of previous arguments, showing what was really important about the updates)

**Weaknesses:**

Presentation.

**Questions:**

So what $\phi$ should we choose and how to set the step-size? It is not very clear we gained too much by the end of the paper other than seeing that there is a whole set of new update rules one could study.

I was surprised that the exact asymptotic rate does not depend on $\gamma$. But I guess $\gamma$ creeps back in through $\Delta$? Perhaps this should be discussed?

---

> ### Author Response · Authors · 2024-11-17
> **Response to Reviewer 7X97**
>
> Thank you for the detailed reading and pointing out the typos. We will fix them in the revision.
>
> **About the relation to the escort transform based work:** When $p=2$, policy gradient under the escort parameterization is an instance of $\phi$-update, which is not true for $p\ne 2$. However, a class of $\phi$-update (namely polynomial update in Appendix F.2) can be implemented under the escort parameterization, which has been briefly discussed in Remark F.1.
>
> **About the parameterization and prove techniques:** Thank you for suggesting the formulation of $\phi$-update under softmax parameterization, which does provide an interesting intuition for condition (III).\
> Even though $\phi$-update (with any $\phi$) can be implemented under the softmax parameterization, for the global convergence, we note that the proof carried out in the parameter space in [1] still cannot be applied to $\phi$-update with such softmax policy parameterizations. One of the key ingredients in establishing the global convergence of softmax PG in [1] is the property of $\sum_a\theta^k_{s,a}=\sum_{a}\theta^0_{s,a},\forall k$. It comes from the fact of $\sum_a\frac{\partial V^\pi(s)}{\partial\theta_{s,a}}=0$ and the parameter update form of $\theta^{k+1}_{s,a}=\theta^k\_{s,a}+\eta\frac{\partial V^k(s)}{\partial \theta^k\_{s,a}}$. In general, such property is not satisfied for $\phi$-update under the same softmax parameterization. Instead, our proof is completely conducted in the policy space. Thank you again for raising this helpful question, and we may add a remark after the global convergence result.\
> The other convergence results (such as policy convergence and exact asymptotic convergence rate) are new, to the best of our knowledge. Currently, we are now aware  whether they can be proved in the parameter space or not.
>
> **About the global linear convergence:** We agree that the global linear convergence result is not neat enough since the rate $\rho(\varepsilon)$ is the maximum of two rates from two phases of the iterations. That being said, we  still think they are interesting to some extend. For instance, in optimization, global linear convergence usually can only be established for strongly-convex (or strongly-concave) functions, with the convergence rate relying on the parameter controlling the strongly convex property. For problems without this property, it is typically difficult to establish the global linear convergence.  However, we show that for the RL problem (even non-convex and non-concave), the establishment of linear convergence is possible  by leveraging the particular MDP structure even though it is difficult to give a concise expression for the convergence rate due to the lack of certain uniform property. It is worth noting that most of the existing global linear convergence results are either based on entropy regularization or using exponentially increasing step sizes.
>
> **About how to choose $\phi$ and the step-size:** In Appendix F.1, we provide some variants of softmax NPG, which show $p<q$ is preferred since the ''stretch intensity'' around $t=0$ becomes stronger as $p/q$ is smaller. The experiment presented in Figure 4(a) verifies that such $\phi$-update enjoys a better tail convergence than softmax NPG. We also provide a family of polynomial updates in Appendix F.2, and larger $p$ gives a better convergence rate in this case as shown in Figure 4(b). For the step size, Theorem 3.4 implies that larger step size yields a better tail convergence in the exact gradient setting. We have also tentatively conducted some preliminary numerical experiments under neural network parameterization to test the softmax NPG variants, see our general response to all reviewers. Despite this, how to choose $\phi$ and the step size in the practical scenarios  still remains a good direction to explore in the future.
>
> **About the exact convergence rate:** The exact asymptotic rate only explicitly depends on $\Delta$, and it is true that $\Delta$ may rely on $\gamma$. For instance, it can be verified that the $\Delta$ of the Grid World in Figure 2 is $(1-\gamma)\cdot\gamma^{H+W-3}$, where $H$ and $W$ are the height and width of the Grid World.  The fact that the exact asymptotic rate does not rely on $\gamma$ explicitly can be interpreted intuitively as follows. When the policy and value are close to optimal ones, we have $A^k\approx A^*$. In such case, the problem can be roughly viewed as a bandit problem. That is, on each state, we just need to find the actions with largest optimal advantage, i.e. $\arg\max_a\\,A^*(s,a)$. As the bandit problem can be viewed as a MDP with $\gamma=0$, it is natural that the local convergence rate does not (explicitly) depend on $\gamma$. We can give a short discussion after the exact asymptotic rate result.
>
> **References:**
>
> [1] Agarwal, Alekh, et al. *On the theory of policy gradient methods: Optimality, approximation, and distribution shift.* Journal of Machine Learning Research, 2021.

---

> > ### Comment · Reviewer_7X97 · 2024-11-17
> >
> > Thanks for the extra bits of information. Nicely done, good job! I especially like the explanation that in the limit, we get back to bandits. That is a nice way of interpreting what action-values do. I also like (and agree with) the explanation of why previous proofs would not cover your case. It would be nice to expand the paper by adding (at least) these explanations to it.

---

> > > ### Author Response · Authors · 2024-11-18
> > >
> > > Many thanks again for the insightful comments. We have fixed some typos and these explanations have been added as Remark 3.1 and 3.4.

---

### Official Review · Reviewer_29J3 · 2024-11-04

**Soundness:** 3
**Presentation:** 3
**Contribution:** 3
**Rating:** 6
**Confidence:** 3

**Summary:**

This paper studies policy parameterization for tabular MDPs using a scaling function  $\phi$, which serves as a systematic generalization of the softmax NPG method. Convergence rate proofs are provided and numerical simulations are illustrated to validate the theoretical claims.

**Strengths:**

The paper addresses a gap in the literature by extending policy parameterizations of tabular MDPs beyond traditional methods like softmax and escort transformations. By applying a general scaling function  $\phi $ in the policy update step, it potentially contributes to the broader understanding of policy gradient methods analyzed through the mirror descent framework given more discussion of distinction is provided.

**Weaknesses:**

- While the paper claims to offer a novel analysis of general policy updates without relying on standard regularization methods such as softmax NPG, it does not clearly distinguish its approach from existing methods. Given that softmax NPG is equivalent to policy mirror descent with a Kullback-Leibler (KL) divergence regularizer (Shani et al., 2020), the paper needs to clearly explain how the proposed  $\phi $-update differs from or improves upon established mirror descent techniques. Without this distinction, the originality and necessity of the proposed analysis approach remain unclear if current theory literature can already encapsulate such approaches with a general regularize used in the mirror descent policy update step (Lan 2023).

- The paper briefly mentions the application of the proposed method in function approximation settings where  $\phi$  is induced by a neural network in Section 4.2. However, it lacks a concrete implementation example or discussion on how the necessary assumptions can be verified in practice. Including such an implementation would enhance the practical relevance and demonstrate the method’s applicability despite potentially unverifiable assumptions.

Shani, Lior, Yonathan Efroni, and Shie Mannor. “Adaptive Trust Region Policy Optimization: Global Convergence and Faster Rates for Regularized MDPs.” Proceedings of the AAAI Conference on Artificial Intelligence, vol. 34, no. 04, 2020.

Lan, Guanghui. "Policy mirror descent for reinforcement learning: Linear convergence, new sampling complexity, and generalized problem classes." Mathematical programming 198.1 (2023): 1059-1106.

**Questions:**

In the context of implementing general policy parameterizations with neural networks (Section 4.2), how can the underlying assumptions of the proposed method be verified or satisfied? Specifically, is the approach compatible with input-convex neural networks (or other popular parametrization scheme for finite MDP with function approximation), and if so, how does it ensure the validity of the theoretical guarantees presented?

---

> ### Author Response · Authors · 2024-11-17
> **Response to Reviewer 29J3**
>
> **About the proof novelty:** Even though softmax NPG can be viewed as an instance of PMD with KL divergence, for a general $\phi$, it is difficult to find a corresponding Bregman divergence such that $\phi$-update can fit into the PMD framework, for example consider the sigmoid function. Therefore, the existing analysis for PMD does not apply for general $\phi$-update. Instead we carry out the proofs by computing the Bellman improvement and establishing the contraction of the probability of non-optimal actions, which are substantially different from the analysis based on smoothness and gradient domination (e.g. softmax PG in [1]) or three point lemma (e.g. PMD in [2]). Based on our analysis, we have established two totally new convergence results: policy convergence (without regularization) and exact asymptotic convergence rate.
>
> **About the general parameterization and its implementation:** First it should be pointed out that $\phi$ is a scaling function on advantage functions. We didn't mean that $\phi$ is induced by a neural network in Section 4.2, but that the policy can be parameterized by a neural network. The assumptions on $\phi$ can be easily verified and there are a broad class of functions that satisfy these conditions. \
> It is worth pointing out that even focusing on the tabular and the exact gradient settings in this paper, there are still  convergence results which have not appeared previously, i.e. the policy convergence and exact aysmptotic convergence. It is not clear how to extend the analysis  to general policy parameterizations or neural networks, which is too general for us to answer currently. To study this question, we may start from for example log-linear policies or some non-degenerate policies in the future. \
> As demonstrated in Section 4.2, $\phi$-update can be implemented together with neural networks. We have tentatively conducted some preliminary numerical experiments, and results are presented in the table in our general response to all reviewers above. However, comprehensive evaluations of the empirical performance of $\phi$-update is beyond the scope of this paper. Thus we prefer to focus on the theory in this paper, and not to include related numerical results.
>
> **References:**
>
> [1] Mei, Jincheng, et al. *On the global convergence rates of softmax policy gradient methods.* International Conference on Machine Learning, 2020.
>
> [2] Lan, Guanghui. *Policy mirror descent for reinforcement learning: Linear convergence, new sampling complexity, and generalized problem classes.* Mathematical Programming, 2021.

---

> ### Author Response · Authors · 2024-11-28
>
> Dear Reviewer 29J3:
>
> Since the discussion period has been extended, we are wondering whether there are further issues you have for us to address.
>
> Best regards,
>
> Authors

---

### Official Review · Reviewer_rtXi · 2024-11-04

**Soundness:** 3
**Presentation:** 3
**Contribution:** 3
**Rating:** 6
**Confidence:** 4

**Summary:**

This paper studies the convergence of $\phi$-Update, which modifies the exp in softmax of natural policy gradient method to another $\phi$ function, which is positive and monotonically increasing.

The authors studies several convergence results with exact gradient updates, including asymptotic convergence, rate of convergence, and policy convergence, and also verified the results using simulations.

**Strengths:**

The proposed $\phi$-update is sort of different but connected with existing methods.

The analysis is comprehensive and theoretical results are fine.

The paper is well written and easy to follow.

**Weaknesses:**

I am a bit concerned with the exact gradient setting. Although it is a slightly different update than softmax PG/NPG variants, it is also fair to say that the exact PG is a well understood and studied setting since Agarwal2019 and many papers after that as noted in the paper.

**Questions:**

I understand that this is mainly theoretical work. However, one suggestion might be that some promising experimental results might show that there is a real motivation of using alternative $\phi$ transforms than standard softmax (also comparisons with existing alternatives should be included).

In terms of extending the work to neural networks, I am a bit concerned this $\phi$ transform is not easy to be extended. The proposed update is multiplicative in policy space (Line 100), and to make that compatible with NNs (say there are logits outputed for actions, and the some transforms of logits will give probabilities over actions), the $\phi$ transform needs to satisfy $\phi(a) \cdot \phi(b) = \phi(a+b)$, which  reduces back to the exp function (softmax transform). Given this, it is hard to see how the $\phi$ transform will be a very promising direction to study in function approximation settings.

---

> ### Author Response · Authors · 2024-11-17
> **Response to Reviewer rtXi**
>
> **About the exact gradient setting:** Though the convergence of policy gradient methods with access to exact policy have been extensively studied in the last few years, there are still fundamental problems to be addressed. For instance, the convergence of policy still remains unclear for most of the policy gradient methods. To the best of our knowledge, the only policy convergence result was established in [1] for a homotopic PMD method based on entropy regularization. In contrast, we first show that the policy produced by $\phi$-update converges even without regularization (Theorem 3.3), which indeed implies the policy convergence of the classical softmax NPG method. Moreover, the exact asymptotic convergence rate (Theorem 3.4) has not appeared previously as far as we know. In order to establish these new results, new proof techniques have been developed. For instance, to establish the policy convergence result, we bound the policy ratio on optimal actions and utilize the local linear convergence result to show that $\\{ \log \pi^k(a^*|s) \\}, a^* \in \mathcal{A}^*_s$ is a Cauchy sequence.
>
> **About the motivation of using alternative $\phi$ transforms and the experimental results:** In Appendix F.1, we have indeed provided some variants of softmax NPG, which show  $p < q$  is preferred since the ''stretch intensity'' around $t=0$ becomes stronger as $p/q$ is smaller. The experiment presented in Figure 4(a) verifies that such $\phi$-update enjoys a better tail convergence than softmax NPG. We also provide a family of polynomial updates in Appendix F.2, and larger $p$ gives a better convergence rate in this case as shown in Figure 4(b). \
> We have tentatively conducted some preliminary numerical experiments under parameterizations with neural networks, and results are presented in the table in our general response to all reviewers above. Because there are still a lot work to do to comprehensively evaluate the empirical performance of $\phi$-update (e.g., how to choose $\phi$ for difference problems), we prefer to focus on the theory in this paper, and not to include related numerical results.
>
> **About the concern of the extension to neural networks:** We don't quite understand why the scaling function $\phi$ should be related to the logits output of neural networks, especially how the property $\phi(a) \cdot \phi(b) = \phi(a+b)$ is obtained. For example, let  the logits  output by neural network at $(s,a)$ be $f_\theta(s,a)$, where $\theta$ is the parameter of the neural network. Then the neural network policy is
> $$ \pi_\theta(a|s) = \frac{\exp(f_\theta(s,a))}{\sum_{a^\prime\in\mathcal{A}}\exp(f_\theta(s,a^\prime))}. $$
> Given current parameter $\theta_k$, to implement $\phi$-update, one can seek a new parameter $\theta_{k+1}$ such that
> $$ \forall \\, s\in\mathcal{S}, \\, a\in\mathcal{A}: \quad f_{\theta_{k+1}}(s,a) = f_{\theta_k}(s,a) + \log(\phi(\eta_k A^{\pi_k}(s,a))). $$
> Since the above equality might not be achieved, we propose to consider the following update in Section 4.2:
> $$ \theta_{k+1} \in \underset{\theta}{\arg\min} \\; \Bbb{E} [\\,\mathrm{KL}(\pi_\theta(\cdot|s) \\, || \\, \pi^k_\phi(\cdot|s))\\,], $$
> where $\pi_\phi^k$ is policy obtained by applying one step of $\phi$-update on $\pi^k$.
>
> **References:**
>
> [1] Li, Yan, et al. *Homotopic policy mirror descent: Policy convergence, algorithmic regularization, and improved sample complexity.* Mathematical Programming, 2023.

---

> > ### Comment · Reviewer_rtXi · 2024-12-01
> > **thank you for the feedback**
> >
> > Thank you for the responses, which addressed my concerns. I will raise my score to a 6.

---

> > > ### Author Response · Authors · 2024-12-02
> > >
> > > Thank you for the positive score.
> > >
> > > Best regards,
> > >
> > > Authors

---

> ### Author Response · Authors · 2024-11-28
>
> Dear Reviewer rtXi:
>
> Since the discussion period has been extended, we are wondering whether there are further issues you have for us to address.
>
> Best regards,
>
> Authors

---

### Author Response · Authors · 2024-11-17
**General Response to All Reviewers**

We want to thank all the reviewers for their time and helpful comments. In this paper, we have presented several new theoretical results for a general policy update rule called $\phi$-update, especially the policy convergence and the exact asymtotic convergence rate, which are also applicable  and new for the classical softmax NPG method.

In this general response, we would like to point out that $\phi$-update can be implemented with general parameterization including neural networks, as sketched in Section~4.2. We have tentatively conducted some preliminary numerical experiments (for the exponential family presented in Section F.1 combined with artificial neural networks) on several environments from MuJoCo, and results are presented in the following table. For each experiment, we compute the mean and the standard deviation of the final accumulative reward accross 10 different random seeds. The policy neural network is a two-layers MLP with 64 units per layer, and the timesteps are 5 millions for each experiment.

Note that the numerical results here only serve as an illustration, are not used to claim the SOTA results of $\phi$-update. In fact, comprehensive empirical evaluations  of $\phi$-update is beyond the
scope of this paper, especially on how to choose $\phi$ for different problems. Thus we focus on the theory in this paper, and intend not to include these preliminary results.


|    | HalfCheetah-v2 | $~~~$Hopper-v2 | Walker2d-v2 |
|:-----:|:-----:|:-----:|:-----:|
| PPO | $3184(\pm 65)$ | $3010(\pm1106)$ | $\mathbf{3041(\pm881)}$ |
| Exp($p=1, q=5, \delta=0.5$) | $\mathbf{3323(\pm484)}$ | $\mathbf{3607(\pm1217)}$ | $2598(\pm845)$ |
| Softmax NPG (i.e. Exp($p=1, q=1$)) | $2764\pm(129)$ | $3362(\pm1335)$ | $2047(\pm913)$ |
| Exp($p=5, q=3$) | $2262(\pm351)$ | $1888(\pm830)$ | $1246(\pm470)$ |

---

### Author Response · Authors · 2024-11-22

As suggested by reviewer NGsf, we have included the preliminary numerical results  (in terms of plots instead of table for more clarity)
on $\phi$-update under the neural network parameterization in Appendix G.

---

### Meta-Review · Area_Chair_6TT9 · 2024-12-07

**Metareview:**

This paper considers a general policy update framework for discounted MDP, called $\phi$-update, which includes but not limited to several popular policy gradient updates such as softmax PG, softmax NPG and and Hardamard PG. In this framework, the new $\pi(a|s)$ takes a multiplicative form of the old $\pi(a|s)$ and a scaling factor of the advantage function. The authors proposed a set of conditions on the scaling function under which desirable convergence results can be obtained, which provides a common framework for the convergence analysis of a class of policy updating schemes. Moreover, as the update scheme goes beyond the policy gradient descent (or mirror descent) framework, it provides us a deeper understanding of how the current policy optimization method works. Based on the contribution of the paper, we decide to accept the paper.

**Additional Comments On Reviewer Discussion:**

Besides the minor concerns that have been well-addressed by the authors, a major change of this paper is that the authors have added a section of preliminary numerical experiments.

---

### Decision · Program_Chairs · 2025-01-22

Accept (Poster)